



# A local model of snow-firn dynamics and application to Colle Gnifetti site

Fabiola Banfi[1] and Carlo De Michele[1]

[1]Department of Civil and Environmental Engineering, Politecnico di Milano, Milano, Italy

**Correspondence:** Fabiola Banfi (fabiola.banfi@polimi.it), Carlo De Michele (carlo.demichele@polimi.it)

**Abstract.** The regulating role of glaciers on catchment run-off is of fundamental importance in sustaining people living in low lying areas. The reduction in glacierized areas under the effect of climate change disrupts the distribution and amount of run-off, threatening water supply, agriculture and hydropower. The prediction of these changes requires models that integrate hydrological, nivological and glaciological processes. In this work we propose a local model that combines the nivological and glaciological scales. The model describes the formation and evolution of the snowpack and the firn below it, under the influence of temperature, wind speed and precipitation. The model has been implemented in two versions: (1) a multi-layer one that considers separately each firn layer, and (2) a single-layer one that models firn and underlying glacier ice as a single layer. The model was applied at the site of Colle Gnifetti (Monte Rosa massif, 4400–4550 m a.s.l.). We observed an average reduction of annual snow accumulation due to wind erosion of $2 \cdot 10^3 \mathrm{\, kg \, m^{-2} \, y^{-1}}$ to be compared with a mean annual precipitation of about $2.7 \cdot 10^3 \mathrm{\, kg \, m^{-2} \, y^{-1}}$. The conserved accumulation is made up mainly of snow deposited between April and September, when temperatures above melting point are also observed. End of year snow density, instead, increased in average of $65 \mathrm{\, kg \, m^{-3}}$ when the contribution of wind to snow compaction was added. Observations show a high spatial and interannual variability in the characteristics of snow and firn at the site and a correlation of net balance with radiation and number of melt layers. The computation of snowmelt in the model as a solely function of air temperature may therefore be one of the reasons of the observed mismatch between model and observations.

## 1 Introduction

Glacier ice covers almost 16 million $\mathrm{km^2}$ of the Earth's surface, of which it is estimated that only 3 % is retained by the mountains outside the polar region (Benn and Evans, 2010). Despite this small percentage the amount of water stored in mountain glaciers plays a key role in sustaining people living in low lying areas (Adhikary, 1993), influencing run-off on a wide range of temporal and spatial scales (Jansson et al., 2003; Huss et al., 2010). Storing water coming from precipitation in winter and delaying the time in which it reaches the river network, they sustain streamflow in hotter and drier periods when precipitation is lacking and when it is most needed for agriculture and as drinking water (Fountain and Tangborn, 1985; Hagg et al., 2007). Jost et al. (2012) found in upper Columbia river basin (Canada) covered for only 5 % by glaciers, that ice melt contributes up to 25 % and 35 % to streamflow respectively in August and September and between 3 % and 9 % to total streamflow. In high mountain river basins of the northern Tien Shan (Central Asia), with areas of glaciation higher than



30–40 %, glaciermelt contribution is 18–28 % of annual run-off but it can increase to 40–70 % during summer (Aizen et al., 1996).

The reduction of glacier volume observed over the past 150 years (Vaughan et al., 2013; Hock et al., 2019) will result in a change in the present distribution and amount of water storage and release with implications in all aspects of watershed
management (Hock et al., 2005) with consequent high economic impacts (Huss et al., 2010). The prediction of these changes is therefore fundamental in order to asses and reduce their impacts, optimizing consequently the management of water resources. To accomplish this task, models that integrate hydrological, nivological and glaciological components and that consider a variable glacier extension and the transient response of glacier to climate change are required (Luo et al., 2013).

Despite their importance, fully integrated glacio-hydrological catchment models are not common in literature (Wortmann
et al., 2019). Some examples of glacio-hydrological models are provided by the works of Huss et al. (2010); Naz et al. (2014); Seibert et al. (2018) and Wortmann et al. (2019).

Wortmann et al. (2019) grouped the main problems of glacio-hydrological models in two categories: integration and scale. With integration problems they refer to the simplified or absent description of the remaining catchment hydrology in models that describe in detail glacier processes. The decrease in the fraction of ice covered areas requires a proper description of
both components also in basins that in the present are highly glacierized. Another aspect is the integration of nivological and glaciological components: a joint simulation of glacier mass balance and snow accumulation and melt is required in order to avoid inconsistencies (Jost et al., 2012; Naz et al., 2014). The problems of scale arise from the different resolutions required by glacial, nivological and hydrological processes. Physically based models that consider all glacier processes (mass balance, subglacial drainage and ice flow dynamics) are often too computationally expensive to be used in a combined glacio-
hydrological model that considers the entire catchment. In addition they are characterized by a complexity higher than the one of many semi-distributed hydrological models. It is therefore necessary to develop glacier models with a degree of complexity similar to the one of hydrological models but that are still able to reproduce the important processes (Seibert et al., 2018).

In the present work we give our contribution proposing a local model that follows the transformation of snow into firn and glacier ice under the influence of temperature, precipitation and wind speed. The equations of the model were derived from
mass balance, momentum balance and rheological equations and they estimate the snowpack and firn characteristics (depth and density of snow and firn, depth of water and refrozen meltwater and rain inside the snowpack). We present two versions of the model: (1) a version (multi-layer) that considers separately each firn layer, and (2) a version (single-layer) that models firn and underlying glacier ice as a single layer. The latter consists of only six equations and it is therefore more suitable for a possible application in a hydrological model. The former consists of four equations for the snowpack plus two equations for each firn
layer. Providing a profile of density with depth, it gives more insight on the influence of meteorological variables on snow and firn characteristics. Besides, it allows a better validation of the snow-firn model. The equations that describe the snowpack are derived from the work of De Michele et al. (2013) and later Avanzi et al. (2015), modified in order to take into account the contribution of wind erosion and the transformation of snow into firn. To model the firn component, both the densification model of Arnaud et al. (2000) and the one of Herron and Langway (1980) were implemented. In order to test the model a high
altitude site, Colle Gnifetti, belonging to the Monte Rosa massif was chosen.



The manuscript is organized in the following way: we provide the model in Sec. 2; illustrate the case study in Sec. 3; give the results in Sec. 4 and discussion in Sec. 5. The conclusions are given in Sec. 6.

## 2 Methodology

In this section, firstly the snowpack model, proposed by De Michele et al. (2013) and later modified by Avanzi et al. (2015), with the addition of the contribution of wind to snow transport is illustrated and secondly the model with the integration of snow and firn processes is presented.

### 2.1 Snow model

The snowpack is modelled, according to De Michele et al. (2013) and Avanzi et al. (2015), as a mixture of dry and wet constituents. The solid deformable skeleton, that consists of both snow grains and pores, has a total volume $V_S$, unit area, height $h_S$, mass $M_S$ and density $\rho_S$. The liquid water inside the pores has a volume $V_W$, unit area, height $h_W$, mass $M_W$ and constant density $\rho_W = 1000 \ \mathrm{kg \, m^{-3}}$. The refrozen meltwater and rain inside the pores has a volume $V_{MF}$ with unit area, height $h_{MF}$, mass $M_{MF}$ and constant density $\rho_i = 917 \ \mathrm{kg \, m^{-3}}$. It is also possible to define the bulk snow density ($\rho$), snow water equivalent ($SWE$) and volumetric liquid water content ($\theta_W$) as $\rho = (\rho_S h_S + \rho_W h_W + \rho_i h_{MF})/h$, $SWE = (\rho h)/\rho_W$ and $\theta_W = h_W/h$ where $h$ is the height of the snowpack equal to $h = h_S + \langle h_{MF} + h_W - \phi h_S \rangle$ (Avanzi et al., 2015) in which $\phi$ is the porosity and $\langle \rangle$ are the Macaulay brackets that provide zero when the argument is negative and its value when it is positive. The height $h$ and $h_S$ always coincide except at the end of the snowpack existence when the liquid part and the solid part due to refreezing become predominant (i.e. $h_{MF} + h_W > \phi h_S$). In this case, $h > h_S$ because a layer of water and/or ice forms on top of the deformable skeleton.

The model solves the mass balance for the dry and liquid mass of the snowpack and the momentum balance and rheological equation for the solid deformable skeleton, resulting in four Ordinary Differential Equations (ODEs) in the variables $h_S$, $h_W$, $h_{MF}$ and $\rho_S$. The mass fluxes considered are (1) solid precipitation events, snow melt and wind erosion for the dry snow mass, (2) rain events, snow melt, melt-freeze inside the snowpack and run-off for the liquid mass and (3) melt-freeze for the mass of ice. The dry snow density is obtained considering (1) the compaction of snow due to the settling and metamorphosis of grains not driven by wind (2) the increase in densification rate due to drifting snow compaction and (3) a densification due to the addition of new mass. Accordingly, the following system is obtained (see Appendix A for the derivation of the system and the





detailed description of the terms in the equations):

$$\frac{dh_S}{dt} = -\frac{h_S}{\rho_S}\frac{d\rho_S}{dt} + \frac{\rho_{NS}}{\rho_S}s - (I \cdot a)(T_A - T_\tau) - \frac{Q}{\rho_S} \tag{1a}$$

$$\frac{dh_W}{dt} = r + \frac{\rho_S}{\rho_W}(I \cdot a)(T_A - T_\tau) + (I^* \cdot e \cdot a)(T_A - T_\tau) - \alpha \cdot K_W \tag{1b}$$

$$\frac{dh_{MF}}{dt} = -\frac{\rho_W}{\rho_i}(I^* \cdot e \cdot a)(T_A - T_\tau) \tag{1c}$$

$$\frac{d\rho_S}{dt} = (c \cdot A_1 \cdot U)\rho_S \exp(-B \cdot (T_\tau - T_S) - A_2 \cdot \rho_S) + \frac{\rho_{NS} - \rho_S}{h_S}s \tag{1d}$$

In Eq. (1a), $\rho_{NS}$ is the density of fresh snow ($\mathrm{kg\,m^{-3}}$), $s$ is the solid precipitation rate ($\mathrm{m\,h^{-1}}$), $a$ is a calibration parameter ($\mathrm{m\,h^{-1}\,°C^{-1}}$), $T_A$ and $T_\tau$ are the air temperature and the threshold temperature for melting (°C), $I$ is equal to $\frac{h_S}{h_S+k}$ with $k = 0.01$ m if $T_A \geq T_\tau$ and zero otherwise (Avanzi et al., 2015) and $Q$ is the mass of snow eroded by wind ($\mathrm{kg\,m^{-2}\,h^{-1}}$). In Eq. (1b), $r$ is the liquid precipitation rate ($\mathrm{m\,h^{-1}}$), $e$ is a calibration parameter, $I^*$ is equal to $\frac{h_W}{h_W+k}$ if $T_A < T_\tau$ and to $\frac{h_{MF}}{h_{MF}+k}$ if $T_A > T_\tau$ (Avanzi et al., 2015), $\alpha = 1.9692 \cdot 10^9$ $\mathrm{m^{-1}\,h^{-1}}$ (DeWalle and Rango, 2008) and $K_W$ is the intrinsic permeability of water in snow ($\mathrm{m^2}$). In Eq. (1d), $c = 0.10 \cdot 3600$ $\mathrm{s\,h^{-1}}$, $A_1 = 0.0013$ $\mathrm{m^{-1}}$, $A_2 = 0.021$ $\mathrm{m^3\,kg^{-1}}$, $B = 0.08$ $\mathrm{K^{-1}}$ (Liston et al., 2007), $U$ is the wind speed contribution ($\mathrm{m\,s^{-1}}$) and $T_S$ is the average snow temperature (°C) obtained assuming thermal equilibrium between the constituents and a bilinear profile of temperature through depth (see De Michele et al. (2013) for further details).

With respect to the model by De Michele et al. (2013) and Avanzi et al. (2015), the version presented in this work includes the effect of wind both on mass balance and densification. This is important when the model is applied to high altitude sites: Haeberli and Alean (1985), in fact, suggested that a major part of the decrease of accumulation with altitude in the Alps, that occurs above about 3500 m a.s.l., may be due to wind effects.

In analogy with solid transport, snow is mobilized only when wind velocity at the surface exceeds a given threshold that depends on physical proprieties of the surface snowpack (Li and Pomeroy, 1997). Once transport begins, snow can travel in two main modes: saltation and suspension (Déry and Taylor, 1996; Pomeroy et al., 1997). The total snow transport $Q$ is computed by the model with the following assumptions: (1) only snow erosion occurs and no deposition of snow eroded in other positions is present; (2) measured wind speed is always referred to 10 m height, i.e. the height of the snow on the ground is neglected; (2) wind cannot erode snow that experienced a temperature greater than 0 °C for the presence of ice crusts or wet layers following Vionnet et al. (2018). These last two assumptions allow to compute the series of total snow transport $Q$ decoupled from the snow model since knowledge of snow height is not required. In order to implement the routine we followed, with some modifications, Lehning et al. (2000), where a model of snowdrift was added to the one-dimensional snow model SNOWPACK (further details about the implementation of the routine are reported in Appendix A).

## 2.2 Model of snow-firn dynamics

We propose here two versions of the snow-firn model. The first version (single-layer) models firn and underlying glacier ice as a single layer (Fig. 1, left panel). The resulting output is an average density and the total column height. The second version





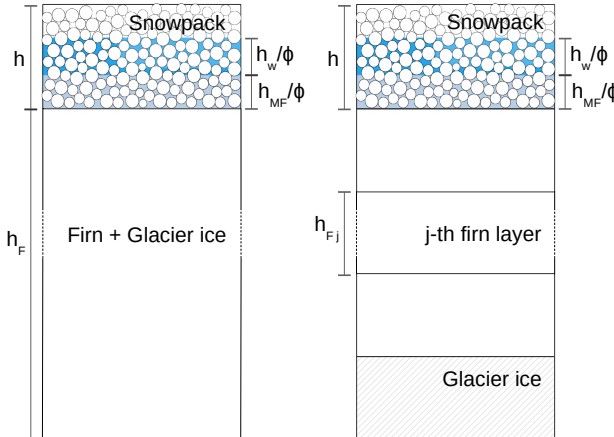

**Figure 1.** A column of snow, firn and ice as modelled by the single-layer (left panel) and multi-layer (right panel) version of the snow-firn model.

(multi-layer) considers separately each firn layer (Fig. 1, right panel) and it allows to distinguish between layers of firn and glacier ice. The discretized profile of density with depth can be obtained from this second implementation.

In both cases we neglected the amount of water percolation inside firn. The presence of water inside firn varies greatly
depending on the type of glacier. At high altitudes, where maximum temperatures are rarely positive, the effects of percolation due to melting are limited (Smiraglia et al., 2000); at the cold site of Colle Gnifetti, where the model was applied, percolation occurs only in the few centimetres below the surface and it does not involve previous year layers (Alean et al., 1983). If needed, the structure of the model allows to easily implement additional processes.

In order to separate snow from firn we refer to its original definition according to which firn is snow that has survived one
melt season (Cuffey and Paterson, 2010).

### 2.2.1  One-layer modelling of firn

The model is composed of two layers: the snowpack (see Sec. 2.1) and the column of firn below it. The firn is modelled as a single impermeable layer of volume $V_F$, unit area, height $h_F$, mass $M_F$ and density $\rho_F$ (Fig. 1, left panel).

The model consists of six ODEs: the four equations of the snow model with in addition the mass balance and momentum
balance of firn. The mass variation of firn is obtained considering firn melt, the effects of precipitation on firn and the transformation of snow in firn at the end of each hydrological year. The firn densification rate is obtained considering a densification due to overburden stress and a densification due to addition of new mass. Accordingly, the resulting system is as follows (see

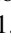



Appendix A for the derivation of the system and the detailed description of the terms in the equations):

$$\frac{dh_S}{dt} = -\frac{h_S}{\rho_S}\frac{d\rho_S}{dt} + \frac{\rho_{NS}}{\rho_S}s - (I \cdot a)(T_A - T_\tau) - \frac{Q}{\rho_S} - \sum_i h_S \delta(t - t_i) \tag{2a}$$

$$\frac{dh_W}{dt} = r + \frac{\rho_S}{\rho_W}(I \cdot a)(T_A - T_\tau) + (I^* \cdot e \cdot a)(T_A - T_\tau) - \alpha \cdot K_W - \sum_i h_W \delta(t - t_i) \tag{2b}$$

$$\frac{dh_{MF}}{dt} = -\frac{\rho_W}{\rho_i}(I^* \cdot e \cdot a)(T_A - T_\tau) - \sum_i h_{MF} \delta(t - t_i) \tag{2c}$$

$$\frac{dh_F}{dt} = -\frac{h_F}{\rho_F}\frac{d\rho_F}{dt} - (I_F \cdot a)(T_A - T_\tau)\delta(h_S) + \frac{\rho_W}{\rho_F}r\delta(h_S)\langle T_\tau - T_A \rangle + \sum_i \frac{\rho}{\rho_F}h\delta(t - t_i) \tag{2d}$$

$$\frac{d\rho_S}{dt} = (c \cdot A_1 \cdot U)\rho_S \exp(-B \cdot (T_\tau - T_S) - A_2 \cdot \rho_S) + \frac{\rho_{NS} - \rho_S}{h_S}s \tag{2e}$$

$$\frac{d\rho_F}{dt} = \frac{d\rho_F}{dt}\Big|_{comp} + \sum_i \frac{\rho - \rho_F}{h_F}h\delta(t - t_i) \tag{2f}$$

The last terms in Eqs. (2a–2c) move, at the end of each melt season, the remaining snowpack (if present) in the firn layer; $t_i$ is the time instant at the end of hydrological year $i$ and $\delta(.)$ is the Dirac delta function. In Eq. (2d), $I_F$ is equal to $\frac{h_F}{h_F + k}$, with $k$ specified above, if $T_A \geq T_\tau$ and zero otherwise. In Eq. (2f), $\frac{d\rho_F}{dt}|_{comp}$ is the densification of firn due to compaction (see Sec. 2.4). Equations (2a–2f) are impulsive differential equations (see e.g., Bainov and Simeonov (1993), for math details). This type of differential equations involving impulse effect are used to describe the evolution of many physical phenomena that have a sudden change in their states such as mechanical systems with impact, biological systems such as heart beats, blood flows, and population dynamics.

## 2.3 Multi-layer modelling of firn

Firn is modelled as a multi-layer column where each layer $j$ has volume $V_{Fj}$, unit area, height $h_{Fj}$, mass $M_{Fj}$ and density $\rho_{Fj}$.





The equations of the model change as follows:

$$\frac{dh_S}{dt} = -\frac{h_S}{\rho_S}\frac{d\rho_S}{dt} + \frac{\rho_{NS}}{\rho_S}s - (I \cdot a)(T_A - T_\tau) - \frac{Q}{\rho_S} - \sum_i h_S \delta(t - t_i) \tag{3a}$$

$$\frac{dh_W}{dt} = r + \frac{\rho_S}{\rho_W}(I \cdot a)(T_A - T_\tau) + (I^* \cdot e \cdot a)(T_A - T_\tau) - \alpha \cdot K_W - \sum_i h_W \delta(t - t_i) \tag{3b}$$

$$\frac{dh_{MF}}{dt} = -\frac{\rho_W}{\rho_i}(I^* \cdot e \cdot a)(T_A - T_\tau) - \sum_i h_{MF} \delta(t - t_i) \tag{3c}$$

$$\frac{dh_{F1}}{dt} = -\frac{h_{F1}}{\rho_{F1}}\frac{d\rho_{F1}}{dt} - (I_F \cdot a)(T_A - T_\tau)\delta(h_S) + \frac{\rho_W}{\rho_{F1}}r\delta(h_S)\langle T_\tau - T_A\rangle + \sum_i \frac{\rho}{\rho_{F1}}h\delta(t - t_i) - \sum_i h_{F1}\delta(t - t_i) \tag{3d}$$

$$\frac{dh_{Fj}}{dt} = -\frac{h_{Fj}}{\rho_{Fj}}\frac{d\rho_{Fj}}{dt} + \sum_i h_{Fj-1}\delta(t - t_i) - \sum_i h_{Fj}\delta(t - t_i) \tag{3e}$$

$$\frac{d\rho_S}{dt} = (c \cdot A_1 \cdot U)\rho_S\exp(-B \cdot (T_\tau - T_S) - A_2 \cdot \rho_S) + \frac{\rho_{NS} - \rho_S}{h_S}s \tag{3f}$$

$$\frac{d\rho_{F1}}{dt} = \frac{d\rho_{F1}}{dt}\Big|_{comp} + \sum_i \frac{\rho - \rho_{F1}}{h_{F1}}h\delta(t - t_i) \tag{3g}$$

$$\frac{d\rho_{Fj}}{dt} = \frac{d\rho_{Fj}}{dt}\Big|_{comp} \tag{3h}$$

where $j$ goes from two to the total number of firn layers. Firn layers that reach the ice density or whose height goes to zero are removed from the model.

## 2.4 Firn densification

The densification of firn due to compaction is usually subdivided into three stages: (1) a first stage dominated by the settling of grains that allows to reach densities up to about $550\,\mathrm{kg\,m^{-3}}$; (2) a second stage dominated by sintering that extends up to the close off density (i.e. the density at which pores become isolated) of about $830\,\mathrm{kg\,m^{-3}}$; (3) a last stage that ends when ice density is reached in which further densification is driven by the compression of the bubbles of air (Cuffey and Paterson, 2010). Different models of firn densification are available in literature (see Lundin et al. (2017) for a review). Here we implemented the model of Arnaud et al. (2000) with some of the modifications proposed by Bréant et al. (2017) (we will refer to it with the acronym AR) and the model of Herron and Langway (1980) (we will refer to it with the acronym HL). Other models could also be implemented. Both HL and AR were developed for polar sites. HL was optimized using ice cores with a mean annual firn temperature between -57 °C and -15 °C and a mean annual accumulation between $0.022 \cdot 10^3\,\mathrm{kg\,m^{-2}\,y^{-1}}$ and $0.5 \cdot 10^3\,\mathrm{kg\,m^{-2}\,y^{-1}}$ while AL was optimized using cores with a mean annual firn temperature between -57 °C and -19 °C and a mean annual accumulation between $0.022 \cdot 10^3\,\mathrm{kg\,m^{-2}\,y^{-1}}$ and $1.1 \cdot 10^3\,\mathrm{kg\,m^{-2}\,y^{-1}}$. In the model of AR the densification has a more physical description with respect to HL that is a semi-empirical model. Besides, AR represents explicitly stresses while in HL the load is parametrized through annual surface accumulation. The model of HL was already applied for non polar ice cores by Huss (2013), where the model was recalibrated in order to match depth-density profiles of temperate/polythermal firn. In the presented application, the parameters were not recalibrated due to the low mean annual firn temperature (MAFT) and low surface accumulation of the studied site.





In AR all three stages of firn densification are modelled. Equations are as follows:

$$\frac{d\rho_F}{dt}\bigg|_{comp} = \begin{cases} \gamma \frac{\max(P, 10^4 \text{ Pa})}{(\rho_F/\rho_i)^2} \left(1 + \frac{0.5}{6} - \frac{5}{3}\frac{\rho_F}{\rho_i}\right)\rho_i & D_D \leq \rho_F/\rho_i \leq D_0 \\ 5.3 A \cdot \left((\rho_F/\rho_i)^2 D_0\right)^{1/3} \left(\frac{a_c}{\pi}\right)^{1/2} \left(\frac{4\pi \cdot P \cdot \rho_i}{3a_c \cdot Z \cdot \rho_F}\right)^3 \rho_i & D_0 < \rho_F/\rho_i \leq D_c \\ 2A \cdot \frac{\rho_F(1-\rho_F/\rho_i)}{\rho_i(1-(1-\rho_F/\rho_i)^{\frac{1}{3}})^3} \left(\frac{2(P-P_b)}{3}\right)^3 \rho_i & D_c < \rho_F/\rho_i \leq 0.95 \\ \frac{9}{4}A \cdot (1-\rho_F/\rho_i)(P-P_b)\rho_i & \rho_F/\rho_i > 0.95 \end{cases} \quad (4)$$

In the first stage ($D_D \leq \rho_F/\rho_i \leq D_0$), $P$ is the overburden pressure (Pa) and $\gamma = \gamma' \exp\left(-\frac{Q_1}{R_G(T_F+273.15)}\right)$ in which $R_G$ is the gas constant, $Q_1$ an activation energy equal to $48 \cdot 10^3 \text{ J mol}^{-1}$, $T_F$ is the average temperature of firn (°C) and $\gamma'$ a parameter whose value is set in order to have a continuous densification rate between the first and second stage (estimated in Sec. 4.2). $D_D$ is the relative surface density and $D_0$ is the relative density at the transition between the first stage and the second stage. The overburden $P$ was computed as the overburden of the midway point of the firn layer. In the second stage

($D_0 < \rho_F/\rho_i \leq D_c$), $A = A_0 \exp\left(-\frac{Q_2}{R_G(T_F+273.15)}\right)$ with $A_0 = 2.84 \cdot 10^{-11} \text{ Pa}^{-3}\text{h}^{-1}$, $a_c$ is the average contact area, $Z$ is the number of particle contacts (see Appendix A for the expression of $a_c$ and $Z$) and $Q_2$ is an activation energy. The value of $Q_2$ was set to $60 \cdot 10^3 \text{ J mol}^{-1}$, as in the model of Arnaud et al. (2000), since it is the typical activation energy associated with self-diffusion of ice. However, at higher temperature (i.e. higher than $-10$ °C) a higher activation energy may be required to best fit density profiles with firn densification models (Cuffey and Paterson, 2010; Arthern et al., 2010; Jacka and Jun,

1994). A discussion of the thermal variation of the creep parameter and the impact of the different sintering mechanisms on it can be found in Bréant et al. (2017). Lastly, in the third stage ($\rho_F/\rho_i > D_c$), $P_b$ is the pressure inside the bubbles equal to $P_b = P_c \frac{(\rho_F/\rho_i)(1-D_c)}{D_c \cdot (1-\rho_F/\rho_i)}$ with $D_c$ and $P_c$ the relative density and pressure at the transition between second and third stage.

In HL only the first and second densification stages are modelled. The equations are as follows:

$$\frac{d\rho_F}{dt}\bigg|_{comp} = \begin{cases} k_0 \cdot (\omega \cdot 10^3) \cdot (\rho_i - \rho_F) & \rho_D \leq \rho_F \leq 550 \text{ kg m}^{-3} \\ k_1 \cdot (\omega \cdot 10^3)^{0.5} \cdot (\rho_i - \rho_F) & 550 < \rho_F < 800 \text{ kg m}^{-3} \end{cases} \quad (5)$$

where $k_0 = 11 \exp\left(-\frac{10160}{R_G(T_F+273.15)}\right)$, $k_1 = 575 \exp\left(-\frac{21400}{R_G(T_F+273.15)}\right)$ and $\omega$ is the annual snow accumulation ($\text{kg m}^{-2}\text{ y}^{-1}$). In HL the transition density between first and second stage is fixed and equal to $550 \text{ kg m}^{-3}$. In order to run the model of HL in a dynamic way, for each year we computed the annual accumulation averaging the ones modelled between the year of deposition of the firn layer and the year before the one considered, following Stevens et al. (2020).

Equations (4) and (5) were applied for firn densities higher than a density $\rho_D$. For firn densities lower than $\rho_D$ the densifi-

cation equation of snow was adopted but neglecting wind contribution. In this way the passage from snow to firn densification is driven by snow characteristics rather than snow age. This is important, for example, when consistent fresh snow falls over the snowpack at the end of the hydrological year. The density $\rho_D$ corresponds to the average surface density and it can be estimated extrapolating the depth-density profiles of ice cores up to the surface.



### 2.4.1 Temperature profile

The energetic description of the volume was simplified assuming the constituents in thermal equilibrium and assuming a bilinear profile of temperature through depth. Temperature was assumed to vary linearly from surface temperature $T_0$ to the MAFT at the depth $z_M$ at which seasonal variation of temperature is negligible. At depths higher than $z_M$, temperature was kept constant and equal to MAFT. In cold glaciers the value of MAFT is close to the mean annual air temperature (MAAT) when melt water percolation is limited (Suter et al., 2001) while in temperate glaciers it is equal to the melting temperature

(Cuffey and Paterson, 2010). Surface temperature was fixed equal to $T_A$ if $T_A < 0°C$ and zero elsewhere. Already Huss (2013) assumed a bilinear profile of temperature in order to study temperate firn densification, fixing $z_M$ to 5 m since it is the typical penetration of winter air temperature. The temperature profile was then used to compute the average snow and firn temperatures that influence snow and firn densification.

### 2.5 Numerical model

The model was solved using the forward Euler method with a constant step size, $\Delta t$, of one hour. To compute the last terms in Eqs. (1d), (2f) and (3g) also when $h_S$, $h_F$ and $h_{F_1}$ are zero, these terms were calculated, following De Michele et al. (2013), as $\frac{\rho_{NS}(t)-\rho_S}{h_S(t)+s(t)\Delta t}s(t)$, $\frac{\rho(t)-\rho_F}{h_F(t)+h(t)}\frac{h(t)}{\Delta t}$ and $\frac{\rho(t)-\rho_{F_1}}{h_{F_1}(t)+h(t)}\frac{h(t)}{\Delta t}$ .

## 3 Study area and data

### 3.1 Study area

The site of Colle Gnifetti (CG) is part of the summit ranges of the Monte Rosa massif, Swiss/Italian Alps. It is the uppermost part of the accumulation area of Grenzgletscher and it forms a saddle that lies between Signalkuppe (4554 m a.s.l.) and Zumsteinspitze (4563 m a.s.l.) at an altitude of 4400–4550 m a.s.l. (Lüthi and Funk, 2000) (Fig. 2). The glacier at Colle Gnifetti has a thickness between 60 and 120 m and a MAFT of -14 °C (Wagenbach et al., 2012). The regime is that of a high altitude site, i.e. nearly persistent sub-zero air temperature, a high precipitation total and high wind speed (Suter et al., 2001).

A mean annual precipitation of $2.7 \cdot 10^3$ kg m$^{-2}$ y$^{-1}$ with an interannual variability of $0.8 \cdot 10^3$ kg m$^{-2}$ y$^{-1}$ (Mariani et al., 2014) was estimated for the period 1961–1993 from a core extracted at upper Grenzgletscher (Eichler et al., 2000).

Even though the site is characterized by high precipitation totals, accumulation in the saddle is considerably lower and highly variable over the glacier surface due to wind erosion, with values ranging from about $0.15 \cdot 10^3$ kg m$^{-2}$ y$^{-1}$ to $1.2 \cdot 10^3$ kg m$^{-2}$ y$^{-1}$ depending on the wind exposure (Alean et al., 1983; Lüthi and Funk, 2000; Licciulli et al., 2020). Alean et al.

(1983) measured the accumulation at CG between 17 August 1980 and 23 July 1982 with a network of 30 stakes. For the period between 14 August 1981 and 23 July 1982 the mass balance was negative in all the stakes due to wind erosion, while the net accumulation of hydrological year 1980–1981 varied between $+0.04 \cdot 10^3$ kg m$^{-2}$ y$^{-1}$ and $+1.18 \cdot 10^3$ kg m$^{-2}$ y$^{-1}$ with the highest values on south facing slopes. This occurs because the enhanced melting and refreezing causes the formation of wet layers and ice crusts and because higher temperatures are associated with a faster densification and both these aspects reduce





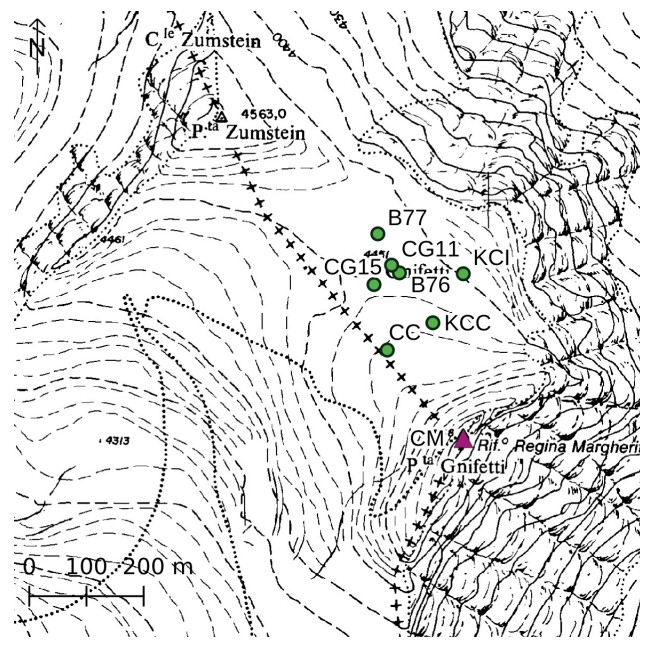

**Figure 2.** The site of Colle Gnifetti and the location of the ice cores considered in the present work. CG03 and CG15 ice core share the same location therefore CG03 is not shown. Source of the basemap: Piedmont Geoportal.

the possibility of wind to erode snow. This results also in the fact that almost all the snow that survives the melt season comes from summer events (Bohleber et al., 2018; Schöner et al., 2002).

### 3.2   Data collection

The stations whose data were used in this study are presented in Fig. 3 and they are summarized in Table 1. Hourly data of air temperature and wind speed at Capanna Regina Margherita (CM) were used as input for the model, hourly data at Passo

del Moro (PM) to reconstruct precipitation at CM, hourly data of air temperature along with daily data at Macugnaga Rifugio Zamboni (MRZ) to calibrate the parameter $a$ and hourly and daily air temperature data at Macugnaga Pecetto (MP), Passo del Moro, Bocchetta delle Pisse (BDP) and Ceppo Morelli (CPM) to infill missing temperature data at Capanna Regina Margherita. Hourly wind speed data at Valtournenche-Cime Bianche (CB) and Col du Grand St Bernard (SB) were used to infill missing wind speed data at CM. All stations belong to Arpa Piemonte with the exclusion of CB and SB that belong respectively to the

Aosta Valley Region and the National Oceanic and Atmospheric Administration (NOAA).

     The station of Capanna Regina Margherita, whose data were used to run the snow-firn model, was installed in 2002 by the Piedmont Region at the Regina Margherita Hut as part of a project that aimed to study the interaction between synoptic flow and orography. With its 4560 m of altitude it can be considered the highest meteorological station in Europe and its wind speed series can be considered representative of the synoptic conditions (Martorina et al., 2003). Due to its recent installation, the use



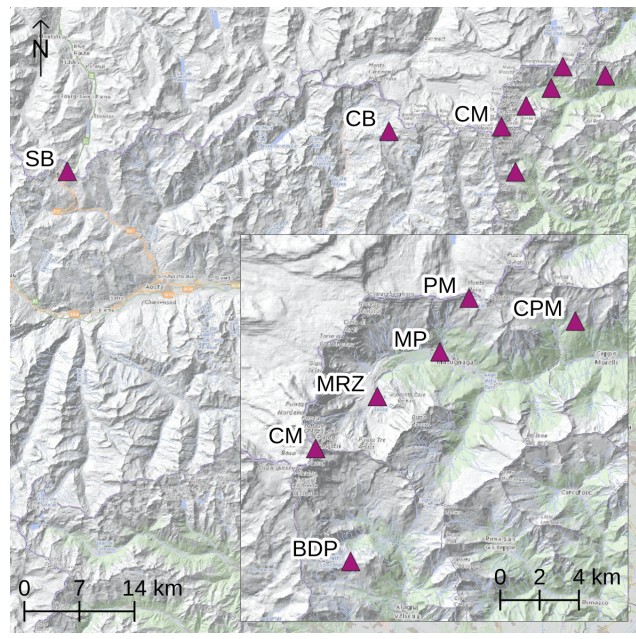

**Figure 3.** Location of the meteorological stations used: Capanna Regina Margherita (CM), Macugnaga Rifugio Zamboni (MRZ), Macugnaga Pecetto (MP), Ceppo Morelli (CPM), Passo del Moro (PM), Bocchetta delle Pisse (BDP), Col du Grand St Bernard (SB) and Valtournenche-Cime Bianche (CB). Source of the basemap: Arpa Piemonte Geoportal.

of these data limits the length of the simulation and the number of cores with which our results can be compared. Nevertheless, we believe that, given the peculiar characteristics of the station, the use of these data may give added value to this study.

In Table 2 ice core data are reported (Fig. 2). Available data consist of some or all of the following information: depth in meters, depth in meters of water equivalent, density and dating. We recall that the first three variables are related, so that one of them can be computed given the other two.

### 3.3 Data handling


The model requires as input a continuous series of air temperature, precipitation and wind speed.

Following the comparison presented by Henn et al. (2013), to fill missing hourly temperature data at Capanna Margherita, the MicroMet preprocessor (Liston and Elder, 2006) was adopted for gap smaller than 24 hours and a long-term lapse rate approach with five stations (CM, MP, CPM, PM, BDP) was adopted for longer gaps. MicroMet is a meteorological model that includes
a data-fill procedure here adopted. The method distinguishes between three conditions: (1) for 1 h gaps the missing datum is replaced with the average of the previous and next measurement; (2) for 2–24 h gaps each missing datum is replaced with the average of the values recorded the next and previous day at the same hour; (3) for longer gaps an auto regressive integrated moving average (ARIMA) model is used. In the period 1 October 2002–13 August 2013, 0.37 % of hourly temperature data were missing. After MicroMet procedure 0.23 % remained missing and were substituted with a long-term lapse rate approach.





**Table 1.** Meteorological data employed in the case study ($p$ stands for precipitation, $SD$ for snow depth, $T_A$ air temperature, $u$ average wind speed and $s$ fresh snow). Hydrological years are identified by the last year, e.g. 2009 is hydrological year 2008–2009. With hydrological year we refer to the period from 1 October to 30 September of the next year.

| Station name | Altitude (m a.s.l) | UTM X WGS84 (m) | UTM Y WGS84 (m) | Variable | Aggregation | Period used | Source |
|---|---|---|---|---|---|---|---|
| Macugnaga Rifugio Zamboni (MRZ) | 2075 | 416068 | 5089214 | $T_A$ | Hourly | Hydrological year: 2009, 2010, 2012, 2015 , 2017, 2018 | Arpa Piemonte |
| Capanna Regina Margherita (CM) | 4560 | 412930 | 5086564 | $T_A, u$ | Hourly | 1 October 2002– 13 August 2013 | Arpa Piemonte |
| Passo del Moro (PM) | 2820 | 420739 | 5094227 | $T_A$ | Daily | 1 October 2002– 30 September 2007 | Arpa Piemonte |
| Passo del Moro (PM) | 2820 | 420739 | 5094227 | $p, SD, T_A, u,$ | Hourly | 1 October 2002, 30 September 2019 | Arpa Piemonte |
| Bocchetta delle Pisse (BDP) | 2410 | 414709 | 5080807 | $T_A$ | Daily | 1 October 2002– 30 September 2007 | Arpa Piemonte |
| Bocchetta delle Pisse (BDP) | 2410 | 414709 | 5080807 | $T_A$ | Hourly | November 2002, September 2007 | Arpa Piemonte |
| Ceppo Morelli (CPM) | 1995 | 426141 | 5093057 | $T_A$ | Daily | 1 October 2002– 30 September 2007 | Arpa Piemonte |
| Ceppo Morelli (CPM) | 1995 | 426141 | 5093057 | $T_A$ | Hourly | November 2002, September 2007 | Arpa Piemonte |
| Valtournenche Cime Bianche (CB) | 3100 | 398610 | 5085987 | $u$ | Hourly | 1 October 2003, 30 September 2019 | Aosta Valley Region |
| Col du Grand St Bernard (SB) | 2479 | 357703 | 5080871 | $u$ | Hourly | 1 October 2002, 30 September 2019 | NOAA |

Wind speed data measured at CM are characterized by repeated zero values that are not observed in near stations and that are probably due to the freezing of the anemometer. In the period 1 October 2002–13 August 2013 nearly 30 % of the wind speed data at CM were equal to zero, while 1.3 % were missing. By comparison, in the same period, there were 2 % zero values in SB series. These zero values were therefore considered missing. To fill missing wind speed data at CM, MicroMet procedure was used for gaps smaller than 24 hours. For gaps longer than 24 hours, data were replaced using measurements at

CB or, if wind speed data were missing also at CB, with data measured at SB. In both series, zero wind speed values recorded for more than four consecutive hours were set missing. In order to take into account the different characteristics of the sites we first computed for each of the three stations the mean and standard deviation for each hour of the year and we removed it from





**Table 2.** Ice core data employed in the case study.

| Name | Drilling date | Mean annual accumulation $(10^3 \ \mathrm{kg\,m^{-2}\,y^{-1}})$ | Data source |
|------|--------------|----------------------------------|-------------|
| B76 | 1976 | 0.37 | Gäggeler et al. (1983) |
| B77 | 1977 | 0.32 | Gäggeler et al. (1983) |
| CG03 | 2003 | 0.45 | Sigl et al. (2018) |
| CG15 | 2015 | 0.45 | Sigl et al. (2018) |
| CG11 | 2011 | 0.41 | Ardenghi (2012) |
| CC | 1982 | 0.22 | Licciulli et al. (2020) |
| KCI | 2005 | 0.14 | Licciulli et al. (2020) |
| KCC | 2013 | 0.22 | Licciulli et al. (2020) |

the data. Missing data at CM were first replaced with the corresponding residual value measured at CB (or SB) and then the final value was obtained using the mean and standard deviation estimated at CM. Reconstructed negative wind speed values at

CM were set to zero. Missing wind speed data at CM were set to zero if they were zero at CB (or SB).

The precipitation series at CM was reconstructed using hourly data measured at PM. The station was chosen due to its vicinity to CM and its altitude of 2820 m a.s.l.. Using the formula proposed by Alpert (1986) and considering a bell shaped mountain, we estimated for the Monte Rosa massif an altitude of maximum precipitation of $z_m = 2547$ m. The altitude of maximum precipitation is away from the crest as it is typical for large mountains (Roe, 2005). We therefore expect to have similar

precipitation totals at CM and PM. The precipitation series measured at PM needs to be integrated with snow depth data since the pluviometer undercatches or does not catch solid precipitation events in winter. In order to reconstruct the total precipitation series, we followed the routine presented by Avanzi et al. (2014). Unlike Avanzi et al. (2014), snow depth variations due to temperature oscillations were removed smoothing the snow depth with a moving average whose window size was calibrated. Even though PM has an altitude higher than the estimated altitude of maximum precipitation, we obtained a mean annual

precipitation of about $2 \cdot 10^3 \ \mathrm{kg\,m^{-2}\,y^{-1}}$ for the period 2002–2019, lower than the one estimated at CG by Mariani et al. (2014). We suppose this may be due to wind erosion events; the procedure implemented by Avanzi et al. (2014), in fact, may compensate for snow depth variations due to wind erosion decreasing the estimated solid precipitation. We therefore increased the resulting hourly solid precipitation with a constant factor in order to match the observed mean annual accumulation at CG of $2.7 \cdot 10^3 \ \mathrm{kg\,m^{-2}\,y^{-1}}$. The total precipitation series was then divided between solid and liquid precipitation using a threshold

of $1 \ ^{\circ}\mathrm{C}$ since this is the value generally found in Europe (Jennings et al., 2018).





### 3.4 Model's parameters

Despite the large number of parameters only two parameters require to be calibrated, namely $a$ and $e$ in Eq.s 1a–1c. All other parameters were taken from literature. The parameter $e$ requires data of snow density or snow water equivalent to be calibrated. Since they were not available, its value was set to 0.2, that is the median value obtained by Avanzi et al. (2015) for a Japanese site. The parameter $a$ was calibrated running the snow model without the wind contribution at MRZ with an hourly time step. For each hydrological year in Table 1 (i.e. 2008-2009, 2009-2010, 2011-2012, 2014-2015, 2016-2017 and 2017-2018) we estimated the parameter using least squares on daily snow depth data and minimizing the objective function with a population-evolution-based algorithm, namely SCE-UA (Shuffled Complex Evolution-University of Arizona) (Duan et al., 1992, 1993). To evaluate the model, for each calibrated parameter, we computed the NSE (Nash-Sutcliffe Efficiency), RMSE (Root Mean Square Error) and MBE (Mean Bias Error) between observed and modelled snow depth for all hydrological years except the one used to calibrate the parameter. The median value of $a$ was selected and used in the snow-firn model. In addition, the parameters of the firn densification model chosen may need calibration if applied to sites significantly different from the polar ones. The parameter $\gamma'$, that governs firn densification rate in AR, instead, does not need calibration since it is fixed once the mean accumulation and MAFT are set. Its value is in fact chosen in order to have a continuous densification rate between first and second stage of densification. For each of the available ice cores, with the exception of CG11, we computed the parameter $\gamma'$ running AR in a steady-state condition (Bader, 1954) using the mean accumulation reported in Table 2. The parameter $\gamma'$ was obtained for two different firn temperatures, $T_F = -14\ °C$ and $T_F = -10\ °C$, that cover the observed ice temperatures at CG (Lüthi and Funk, 2000). The surface density was set to $\rho_D = 360\ \mathrm{kg\,m^{-3}}$ for all cores except for CG03 for which we selected a density of $470\ \mathrm{kg\,m^{-3}}$. Surface density is highly variable at CG due to the influence of wind and aspect. Licciulli et al. (2020) assumed a surface density of $300\ \mathrm{kg\,m^{-3}}$ and $360\ \mathrm{kg\,m^{-3}}$. This last value was here adopted. CG03 has a surface density noticeably higher than the rest of the cores considered, for this reason a different surface density was set for this core. The value was estimated from the density profile of CG03 and CG15. Finally, in order to apply AR we set $D_0 = 0.56$ (Bréant et al., 2017), $P_c = 740 \cdot 10^2\ \mathrm{Pa}$ (Lüthi and Funk, 2000) and $D_c = 0.9$ since the precise value is not known at CG (Lüthi and Funk, 2000). The model of AR was run with a slight modification. We used the first stage densification equation up to a relative density of 0.6, but we kept $D_0 = 0.56$ in the second stage densification equation. The latter, in fact, cannot be applied for $D = D_0$ and it gives densification rates tending to infinity for values tending to $D_0$. The other parameters of the snow-firn model that require to be specified are $z_M$ set to 5 m (Haeberli and Funk, 1991) and the grain radius $R$ that influences the threshold wind speed. It is defined as $R = 3/(\rho_i SSA)$ where $SSA$ is the specific surface area in $\mathrm{m^2\,kg^{-1}}$. $SSA$ was computed adopting the parametrization of Domine et al. (2007) for recent snow, $SSA = -16.051 \ln(\rho_S \cdot 10^{-3}) + 7.01$.



## 4   Results

### 4.1   Parameters' estimation

We obtained a median value of the parameter $a$ of $3.84 \cdot 10^{-4}$ m h$^{-1}$ °C$^{-1}$ with an average NSE, RMSE and MBE of 0.71, 0.293 m and –0.0475 m in validation. The value is in the range obtained by Avanzi et al. (2014) for a selection of forty sites within the SNOTEL network with altitudes between 91 and 3389 m a.s.l.. They obtained median values between $1 \cdot 10^{-4}$ m h$^{-1}$ °C$^{-1}$ and $6 \cdot 10^{-4}$ m h$^{-1}$ °C$^{-1}$. They also observed a slight increase of the parameter with altitude. We therefore tested the influence of adopting the same value estimated at MRZ also for CM, despite the higher altitude. Increasing the parameter $a$ to $6 \cdot 10^{-4}$ m h$^{-1}$ °C$^{-1}$ we obtained a median difference in the annual snow accumulations of $0.010 \cdot 10^{3}$ kg m$^{-2}$ y$^{-1}$, with a maximum value of $0.072 \cdot 10^{3}$ kg m$^{-2}$ y$^{-1}$.

The sensitivity to the parameter $e$ was also tested. Since the value of the parameter is limited between 0 and 1, we increased it from 0.2 to 0.8. The resulting absolute differences in the end of year snow densities are always lower than 13 kg m$^{-3}$, with a median value of about 3 kg m$^{-3}$.

### 4.2   Steady-state firn densification

The depth density profiles obtained using the model of AR and HL in a steady-state condition are reported in Fig. 4. Both HL and AR have very good performances when applied to KCC ice core. The worst performance of AR occurs for CG03 with an underestimation of densification rate for all depths. On the contrary HL has a good fit except for the last part of the profile. For the remaining ice cores the models of AR and HL have a good fit up to depths of about 20–30 m, but they underestimate densification rate below it.

### 4.3   Snow accumulation

The annual accumulation obtained from the snow-firn model is reported in Fig. 5 along with the values retrieved from the three available ice cores, the average value of the observations and its 95% confidence interval. The RMSE between the model and the average of the observations is equal to $0.23 \cdot 10^{3}$ kg m$^{-2}$ y$^{-1}$, while the modelled and observed average annual accumulation and standard deviation are reported in Table 3.

In order to better understand the characteristics of the accumulation at CG the monthly box plot of solid precipitation, snow transport, monthly contribution to annual accumulation and number of hours with $T_A > 0$ °C, that in the model correspond to hours with melting, are provided in Fig. 6. Since snow is moved into firn at the end of September and wind is not allowed to erode firn, the fraction of conserved snow of September may be overestimated and the snow transport of October underestimated. We can see that annual accumulation is composed by snow deposited mainly between April and September, with June the month that in average contributes the most. The months in which solid precipitation is conserved are also the months in which temperature goes above the melting point; winter snow, instead, is completely removed.







**Figure 4.** Observed and modelled depth density profiles. Modelled profiles are obtained running Arnaud et al. (2000) (AR) and Herron and Langway (1980) (HL) in a steady-state condition. The numbers 14 and 10 stand for a mean annual firn temperature of -14 °C and -10 °C, respectively.

**Table 3.** Modelled and observed mean ($\mu$) and standard deviation ($\sigma$) of the accumulation rate for the period 2003–2010.

|  | $\mu$ ($10^3$ kg m$^{-2}$ y$^{-1}$) | $\sigma$ ($10^3$ kg m$^{-2}$ y$^{-1}$) |
|---|---|---|
| Model | 0.46 | 0.16 |
| CG11 | 0.41 | 0.09 |
| KCC | 0.31 | 0.09 |
| CG15 | 0.38 | 0.16 |



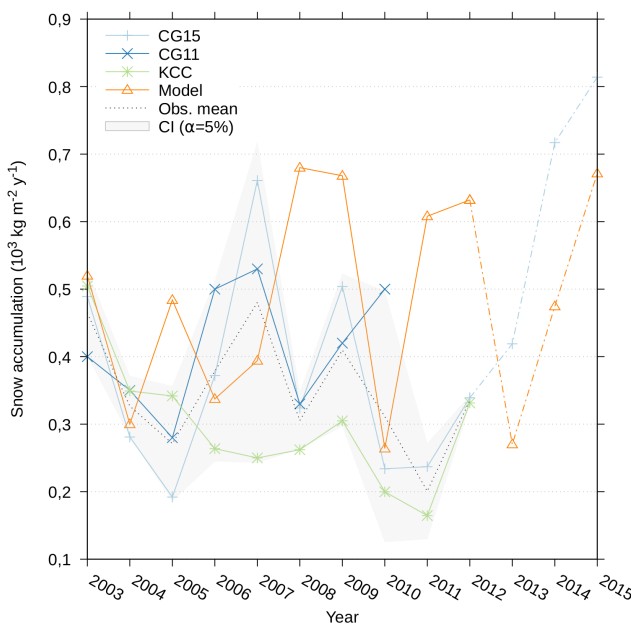

**Figure 5.** Annual accumulation modelled and retrieved from three ice cores. The average of the annual accumulations from ice cores and its 95% confidence interval are also reported.

The influence of wind and temperature on snow accumulation can be seen also in Fig. 7 that reports the scatterplot between the monthly net accumulation divided by the monthly solid precipitation and the median value of wind speed and temperature for the same month. Results are reported only for months with more than four points with non zero net accumulation.

### 4.4    Firn density

### 4.4.1    One-layer model version

The modelled firn density was confronted with the density estimated from KCC and CG15 ice cores (Fig. 8). Since this version of the model returns the average density of all the firn column and not the density of each annual layer, to confront them the model was run for an increasing number of years and the corresponding observed density was obtained averaging the density of the layers deposited in the same range of years. The model was always run up to the drilling date of the two ice cores in order to reproduce the same conditions experienced by the ice core. The results obtained implementing both AR and HL are

reported in Fig. 8. For KCC we fixed the MAFT to -14 °C, while for CG15 to -10 °C, looking for the best performance in Fig. 4.

The model underestimates CG15 average densities while it overestimates KCC average densities. For KCC ice core, the model with AR implemented tends to reach the observed average densities with increasing firn height. In both cases the model with HL implemented results in higher densification rates.



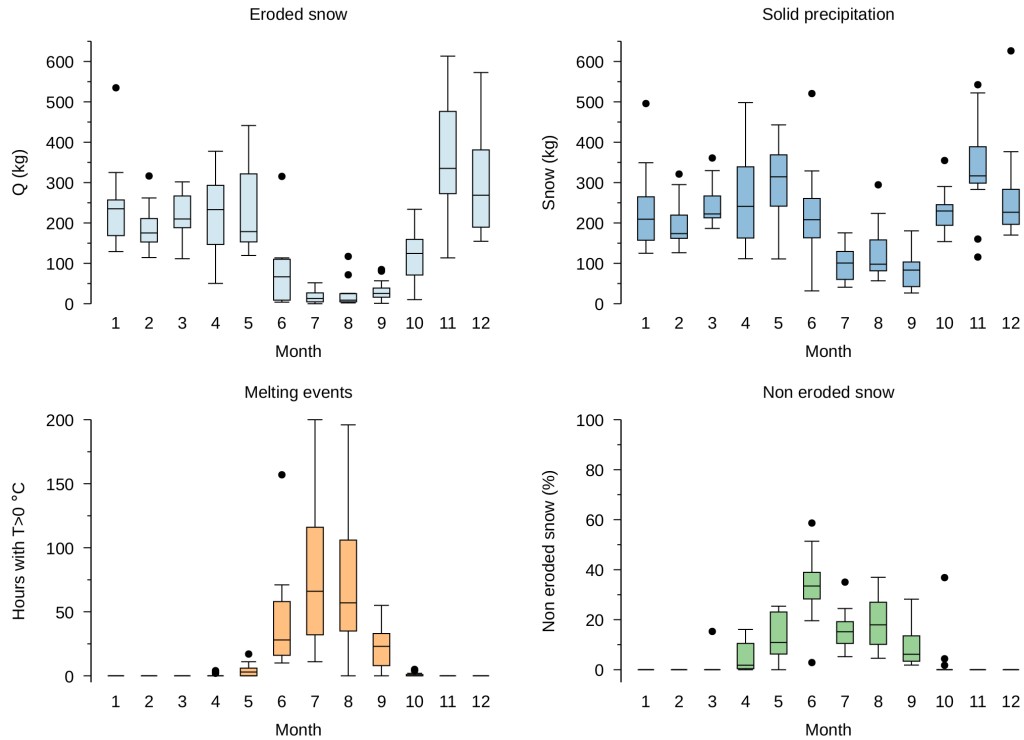

**Figure 6.** Box plot of monthly eroded snow (top left), monthly solid precipitation (top right), monthly number of hours with above zero temperatures (bottom left) and monthly fraction of conserved solid precipitation (bottom right). The horizontal bar inside the box is drawn at the median while the upper and lower ends of the box are drawn at the upper and lower quartile, respectively. The vertical lines, called whiskers, extend up to the most distant point that has a value within 1.5 of the interquartile range. The points outside these limits are drawn individually with dots.

For CG15, the RMSE is $25.3 \, \mathrm{kg \, m^{-3}}$ and $14.1 \, \mathrm{kg \, m^{-3}}$ respectively with AR and HL implemented. For KCC, the RMSE is $20.7 \, \mathrm{kg \, m^{-3}}$ and $46.8 \, \mathrm{kg \, m^{-3}}$ respectively with AR and HL implemented.

### 4.4.2   Multi-layer model version

The modelled density profile was compared with KCC and CG15 density data, implementing in the model both AR and HL (Fig. 9). For CG15, as expected from Fig. 4, the model with AR implemented results in a lower density with respect to the
version with HL. None of the two implementations however reproduce the higher densities of the layers deposited between 2005 and 2007 and between 2013 and 2014. The interannual variability of KCC core is, instead, better reproduce by the two model versions, even though they both overestimate observed densities.

For CG15, the RMSE is $21.6 \, \mathrm{kg \, m^{-3}}$ and $33.4 \, \mathrm{kg \, m^{-3}}$ respectively with AR and HL implemented. For KCC, the RMSE is $38.8 \, \mathrm{kg \, m^{-3}}$ and $58.7 \, \mathrm{kg \, m^{-3}}$ respectively with AR and HL implemented.


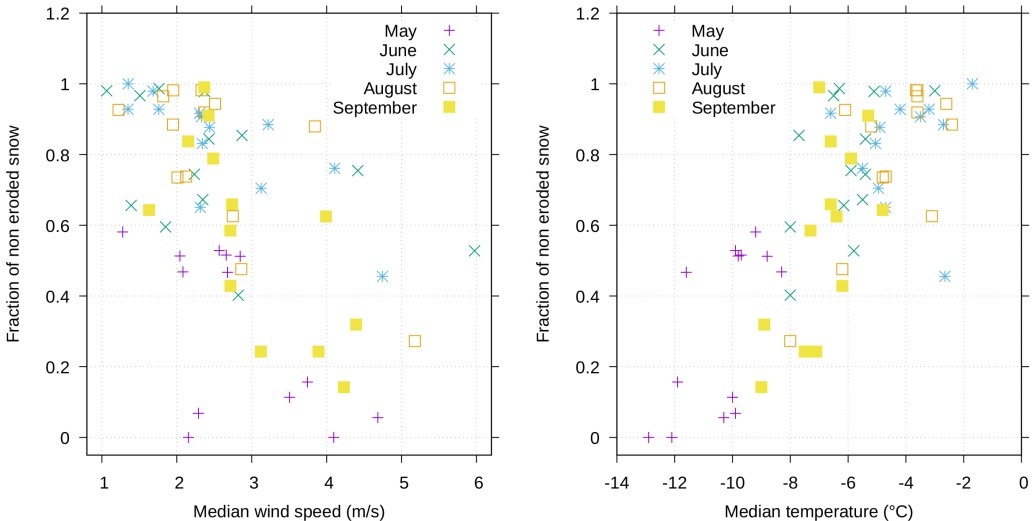

**Figure 7.** Scatterplot between the monthly net accumulation divided by the monthly solid precipitation and the median value of wind speed (left panel) and temperature (right panel) for the same month.

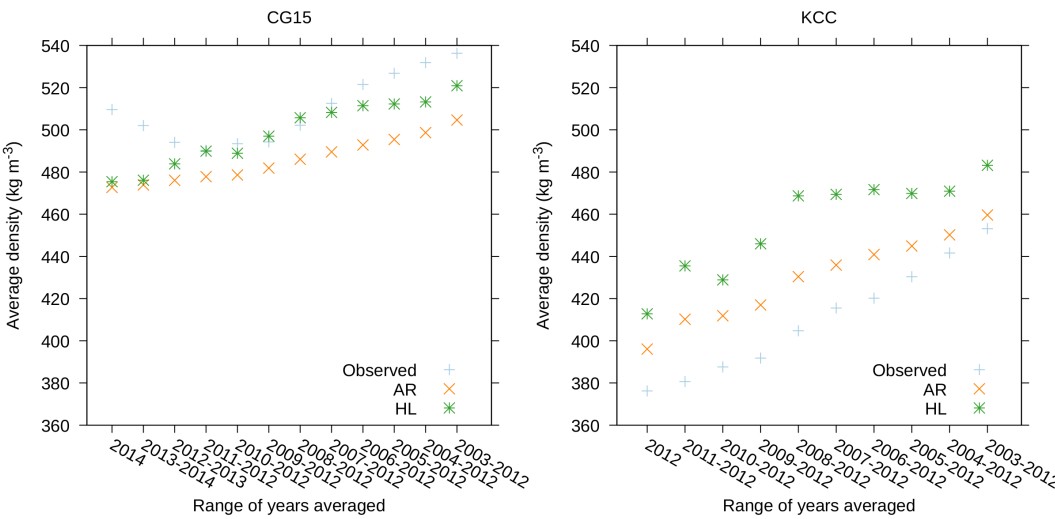

**Figure 8.** Observed and modelled (with one-layer version) average firn density for CG15 (left panel) and KCC (right panel). The range of years in the x-axis indicates the deposition year of the layers whose density was averaged. AR and HL stand for the model version with Arnaud et al. (2000) and Herron and Langway (1980) implemented, respectively.



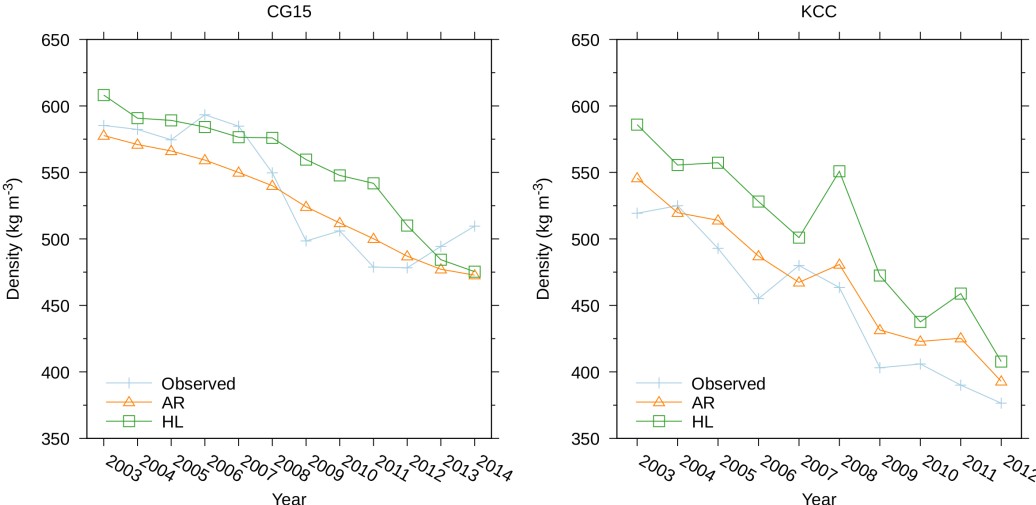

**Figure 9.** Observed and modelled (with multi-layer version) firn density for CG15 (left panel) and KCC (right panel). The x-axis indicates the deposition year of the corresponding layer. AR and HL stand for the model version with Arnaud et al. (2000) and Herron and Langway (1980) implemented, respectively.

## 5 Discussion

### 5.1 Steady-state firn densification

Figure 4 shows a variable performance of the firn densification model depending on the ice core considered; with the exception of KCC and CG15 that show respectively a very good and a very poor performance, for all the other cores we have a good fit up to a density of about 600–700 $\mathrm{kg\,m^{-3}}$. Also Bréant et al. (2017), that modified the original model of AR, observed a variable agreement between data and model, also for sites with similar accumulation and temperature. They suggested that this may be due to different flow regimes of the sites, since their 1D model does not include this effect. Another consideration that emerges from Fig. 4, pointed out also by Bréant et al. (2017), is that the modelled profile results in worse performances when the observed density profile does not show a clear change in densification rate near the critical density $D_0$. The transition is, in fact, more evident for KCC ice core that is associated with the best fit. Finally, Bréant et al. (2017) reported a tendency of the model to overestimate densification rate for lower densities and to underestimate it for higher densities. This is coherent with the results obtained, for which HL predicts lower densities before $D_0$ and higher densities after $D_0$ if compared with AR. In order to compare modelled and observed profiles in Fig. 4 it is important to point out that the two models assume stationary conditions, therefore they are not able to reproduce possible changes in the glaciological characteristics, like mean annual accumulation, and this may lead to a mismatch between data and observations.





## 5.2 Snow accumulation

Snow accumulation at CG is characterized by a high spatial variability (Keck, 2001; Licciulli et al., 2020). The difference in net annual accumulation of CG11 and CG15, that are about 50 m apart, ranges from $+0.13 \cdot 10^3 \ \mathrm{kg\,m^{-2}\,y^{-1}}$ to $-0.266 \cdot 10^3 \ \mathrm{kg\,m^{-2}\,y^{-1}}$ in the period 2002–2012, while the one of CG15 and KCC, that are about 120 m apart, ranges between $+0.41 \cdot 10^3 \ \mathrm{kg\,m^{-2}\,y^{-1}}$ and $-0.15 \cdot 10^3 \ \mathrm{kg\,m^{-2}\,y^{-1}}$. Given the high variability in the accumulation rate, three ice cores may not be enough to fully represent the site, besides, ice core data are biased due to the fact they are drilled preferentially in the north flank, where accumulation is lower. While the modelled average annual accumulation is in the range of the ones estimated for the north flank of CG (Licciulli et al., 2020), the model is not able to reproduce the observed spatial variability. Due to the lack of dependence on topography in the presented model, we do not expect the model to correctly follow one or the other core. The topography influences the amount of solar radiation received that in turn influences melting and wind erosion. At the same time, the topography modifies wind speed that in turn modifies topography itself. This results in a quasi random spatial variation and a systematic temporal variation in surface accumulation at a given location (Keck, 2001). Surface snow temperature was set equal to air temperature, instead of solving the full surface energy balance that would have required a higher availability of data; surface temperatures, in fact, may reach 0 °C also for air temperatures below 0 °C mainly when calm conditions are present or, on the contrary, melting may not occur during positive air temperatures particularly when wind is present (Keck, 2001). In order to have an idea of the sensitivity of the model to the value of the air temperature associated with snow melt, we ran the model with different threshold temperatures for melting (Fig. 10). We recall that this value influences both snow erosion and snow melt, modifying accumulation in two opposite directions. Results show a variable sensitivity to this parameter. Decreasing $T_\tau$ to -1.5 °C results in absolute differences ranging from less than a centimiter up to $0.30 \cdot 10^3 \ \mathrm{kg\,m^{-2}\,y^{-1}}$.

From the box plots in Fig. 6 we observe that the conserved snow is made up mainly by summer precipitation and that the conserved fraction of solid precipitation reflects the number of hours with greater than zero temperature rather than the seasonality of precipitation. The accumulation is, in fact, mainly governed by the wind erosion (Wagenbach et al., 1988) and the presence of wet layers or ice crusts, as well as a faster compaction when temperatures are higher, protects snow from wind erosion. This is well known in ice core studies at CG, since it results in isotope records that are biased towards precipitation of the warm season (Schöner et al., 2002; Bohleber et al., 2013, 2018). On the contrary, it is less acknowledged in other fields. To our knowledge, no models have been applied at CG that try to model the influence of wind speed and temperature on snow accumulation. The only confirmations of this behaviour we have, other than from ice core analysis, come from temporary measurements of the snow height in nearby sites or observations at CG. For example, at Seserjoch (Colle Sesia in Fig. 2), 4300 m a.s.l., the snow height was measured between 1998 and 2000 by Suter et al. (2001) and a main accumulation from about April to November, with practically no accumulation in high winter was observed. Assessing the link between temperature, wind speed and snow accumulation may be important under scenarios of climate warming. This, in fact, could lead, as already suggested by Alean et al. (1983), to a counterintuitive response in a scenario of higher temperatures, since the increased melting


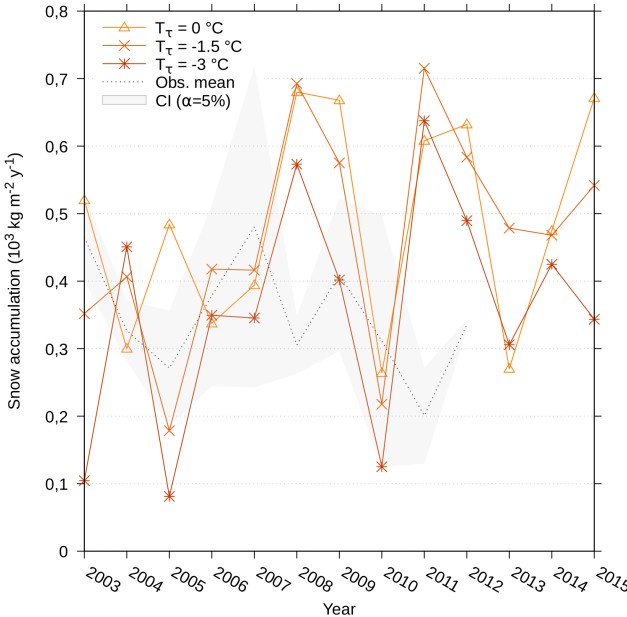

**Figure 10.** Annual accumulation modelled with different values of the threshold temperature for melting $T_\tau$. The average of the annual accumulation of ice core KCC, CG11 and CG15 along with its 95% confidence interval is also reported.

and the reduced fraction of solid events with respect to total precipitation will be accompanied by a reduced snow erosion. An increase of the positive net balances may in turn influence ice avalanche activity.

## 5.3  Firn density

### 5.3.1  One-layer model version

The modelled densities have an opposite behaviour when compared with CG15 or KCC. In the first case, the average density is underestimated while in the second case overestimated. The multi-layer version of the model allows us to better understand the reasons of this mismatch. An error in the density of an individual layer will, in fact, affect all the modelled average firn

densities that contain that individual layer. This is the case of CG15, where the underestimation of the density of the most superficial layers (Fig. 9) influences the average density of the whole firn column. This influence however decreases when more annual layers are averaged. This can be seen for CG15 and the model version with HL implemented, or for KCC and the model version with AR implemented. In the case of CG15 and the model version with AR implemented, underestimation is probably related with the characteristics of the CG15 location. The model of AR, in fact, underestimates CG03 densities (we

recall that CG03 and CG15 are drilled in the same location) also when it is applied alone in a steady-state condition and not inside the presented snow-firn model.

### 5.3.2 Multi-layer model version

The snow-firn model, with both AR and HL implemented, overestimates density when applied to KCC, even though the two firn densification models, applied in a steady-state condition (Fig. 4), provided very good performances with KCC core. This may

be reconducted to the different estimated surface accumulations; the model, in fact, results in an average snow accumulation of $0.46 \cdot 10^3 \ \mathrm{kg \, m^{-2} \, y^{-1}}$ against the $0.22 \cdot 10^3 \ \mathrm{kg \, m^{-2} \, y^{-1}}$ accumulation of KCC. Since the firn densification is driven by overburden stress, this may lead to a systematic bias in the results. The annual behaviour is instead well reproduced. An exception is 2006–2007 hydrological year. This year was characterized by a solid precipitation event that occurred at the end of the hydrological year. As a consequence, the modelled average snow density decreased from $412 \ \mathrm{kg \, m^{-3}}$ at the beginning

of 25 September 2007 to $260.6 \ \mathrm{kg \, m^{-3}}$ at the end of 27 September 2007. The higher annual variability in the modelled density obtained implementing HL is probably related to a higher sensitivity to the surface density, that has instead a low influence on AR as reported also by Bréant et al. (2017).

For CG15, the results of the two different versions of the model are coherent with Fig. 4 with lower densities when AR is implemented and higher densities with HL. In this case, the mean snow accumulations are comparable (i.e. $0.45 \cdot 10^3 \ \mathrm{kg \, m^{-2} \, y^{-1}}$

for CG15 and $0.46 \cdot 10^3 \ \mathrm{kg \, m^{-2} \, y^{-1}}$ for the model). Unlike KCC, the annual variability is not reproduced by either version of the model. The model does not include the spatial variability of solar radiation and wind speed at the site, that causes the two ice cores to have significantly different behaviours. The differences in the parameters of the two model runs are the value of MAFT, $\rho_D$ and $\gamma'$ when AR is implemented. This limits the ability of the model to adapt to the different ice cores. Given the better match with the annual variability of KCC, we may assume that the model better reproduces the conditions at this

location. CG15 is closer to the axis saddle and presumably in a location with a higher exposition to solar radiation since accumulation is doubled with respect to KCC. Consequently, melting events may be more frequent in CG15 ice core and this, along with the snow redistribution due to gravity, could explain the higher surface density. In order to asses the possible influence of an underestimation of melting, we reduced the threshold temperature for snow melt both in wind erosion routine and melt mass flux. Results are reported in Fig. 11. The higher densities of 2006 and 2007 are still not explained by the model, in fact

they are even lower due to lower estimated annual accumulations (Fig. 10). We can instead see an improvement in the ability of the model to follow year-to-year variations in the period 2008–2014.

As already pointed out, in 2006–2007, the average snow density significantly decreased due to a solid precipitation event. This event was recorded not only by the pluviometer at PM, but also at MP and MRZ. The choice of maintaining the densification rate of snow also for firn with a density lower than a given value $\rho_D$ allows to take into account this circumstance.

Due to the low amount of snow that is not eroded, new solid precipitation events, in fact, may have a high influence on snow average density. However, the snow densification equation was applied to firn without including wind effects, even though during winter the most recent firn layer is often not covered by the snowpack. This may lead to underestimation of firn density in hydrological years where a significant mass of new snow is added at the end of the hydrological year. Wind speed has, in fact, a non negligible influence on snow density. The average end of year snow density decreases from $394 \ \mathrm{kg \, m^{-3}}$ to $329 \ \mathrm{kg \, m^{-3}}$

if wind contribution is neglected in fresh snow density and snow densification equations.

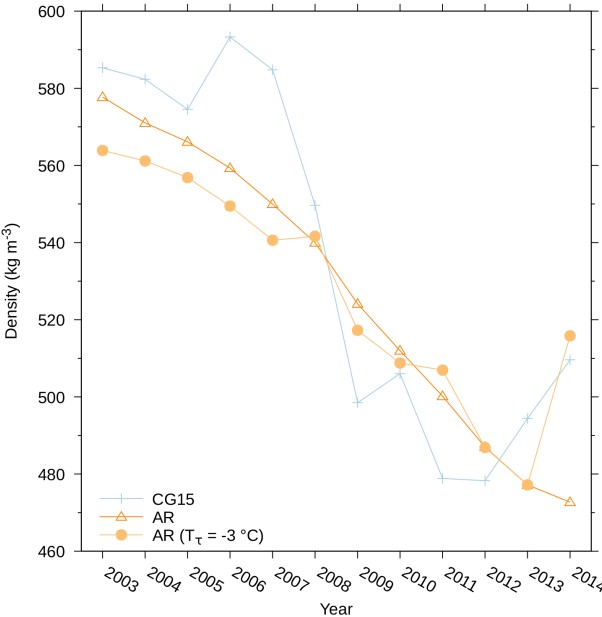

**Figure 11.** Observed and modelled (with multi-layer version) firn density for CG15 ice core. The x-axis indicates the deposition year of the corresponding layer. Results are obtained implementing Arnaud et al. (2000) in the snow-firn model.

A final comment will be given regarding the differences in the results obtained implementing HL or AR in the snow-firn model. The profiles in Fig. 4 show in general a better performance of HL in particular for densities higher than $D_0$. The model of HL was optimized considering also cores with MAFT and accumulation close to the ones of CG, while AR was optimized for cores with lower MAFT. Results in Fig. 9, on the contrary, show a better and more robust performance of the model with AR implemented.

## 6 Conclusions

In this study we have proposed a local modelling that combines snow and firn dynamics and was derived from the mass balance, momentum balance and rheological equations of snow and firn. It requires in input a series of hourly (or sub-hourly) series of precipitation, temperature and wind speed with which the series of snow, water and ice inside the snowpack and firn height along with dry snow and firn density are computed. Two versions of the model were proposed: (1) a version (multi-layer) that considers separately each firn layer, and (2) a version (single-layer) that models firn and underlying glacier ice as a single layer. The two implementations allow to cover different purposes. A simpler model is, in fact, more suitable to reproduce firn inside a hydrological model, where the whole depth-density profile is not necessary and where a reduced number of equations may allow an easier integration. On the other hand, a model that resolves density with depth allows to asses the influence of meteorological variables on snow and firn characteristics. In order to obtain this, we integrated existing firn densification



models into a wider model, therefore moving the boundary of the model from surface accumulation and density to hourly meteorological series. While this may not be needed to retrieve past climatological information from ice cores, it is required to assess the response of the system under present and future changes in the climate.

Both the two model's versions have a parsimonious parametrization, with only two parameters $(a, e)$ to be estimated. In addition, also the parameters of the firn densification equation chosen may need calibration if applied to a temperate site. All the other parameters were, instead, taken from literature. While this choice may allow less flexibility in matching data and observations, it has the advantage not only that the model is less site specific but also that it can be applied to sites with a small amount of data, like most of the high-altitude sites. This was for example the case of CG, where the low availability of data has been a challenge in the application of the model.

In this study, we presented a general modelling that includes all the selected processes, regardless of the presented case study, in order to make it broadly applicable. Then, depending on the specific application, some processes may be neglected with a reduction in the total number of parameters. A high number of parameters are, for example, associated with wind effects that may be neglected in lower altitude sites, but that are likely to be important in high altitude sites (Haeberli and Alean, 1985). At Colle Gnifetti, for example, we saw that observed surface accumulations and densities cannot be explained if wind effects are not considered. The modelled reduction of annual snow accumulation due to wind erosion is on average $2 \cdot 10^3 \ \mathrm{kg\,m^{-2}\,y^{-1}}$ while the increase in end of year snow density is on average $65 \ \mathrm{kg\,m^{-3}}$. The strong wind erosion results also in a greater correlation between the amount of snow preserved in each month and the number of days with above zero temperature rather than with the solid precipitation seasonality. This behaviour may be important in understanding the response of the site to a warming climate. All these elements result in a strong spatial and temporal variability in snow density and accumulation at CG, that is not captured by the model. This would, in fact, require to take in consideration the influence of topography on wind speed and the spatial variation of solar radiation.

The aim of this study was to illustrate the new snow-firn modelling, and to present its potentiality through a case study. In order to integrate it into a hydrological model further steps are required, in particular fluxes among the different columns should be properly considered both regarding runoff and snow transport. In the present modelling, for example, we considered only the snow erosion. At CG the deposition can be neglected due to its characteristics. The site is a saddle with west-east orientation where the predominant wind direction coincides with the saddle orientation. Besides, an ice cliff at the end of it works as a perfect sink for the eroded snow. A distributed modelling needs to take into account the presence of sink and source of snow transport. This will also improve the representation of wind effects on snow accumulation since, in order to include snow transport in a 1D framework, approximation of the process are unavoidable for the nature of the process itself.





## Appendix A: Complete description and derivation of the snow-firn model

### A1 Mass balance equations

The mass balance equations of snow ($M_S$), liquid water in snow ($M_W$), refrozen meltwater and rain in snow ($M_{MF}$), and firn ($M_F$) are as follows:

$$\frac{dM_S}{dt} = P_S - M - Q - E_S \tag{A1a}$$

$$\frac{dM_W}{dt} = P_R + M + F - O - E_W \tag{A1b}$$

$$\frac{dM_{MF}}{dt} = -F - E_{MF} \tag{A1c}$$

$$\frac{dM_F}{dt} = -O_F + P_F + E_S + E_W + E_{MF} \tag{A1d}$$

$P_S$ and $P_R$ are the mass of solid and liquid precipitation events and they are equal to $P_S = s \cdot \rho_{NS}$ and $P_R = r \cdot \rho_W$. The fresh snow density was calculated as proposed by Liston et al. (2007): $\rho_{NS} = \rho_{NS_0} + \rho_{NS_w}$ Following Anderson (1976), $\rho_{NS_0} = 50 \text{ kg m}^{-3}$ if the air temperature $T_A < -15°C$ and $\rho_{NS_0} = 50 + 1.7 \cdot (T_A + 15)^{1.5} \text{ kg m}^{-3}$ otherwise. The second term gives the increase in fresh snow density due to wind and it is computed as $\rho_{NS_w} = D_1 + D_2(1.0 - \exp(-D_3(u_2 - 5)))$ where $u_2$ is the wind speed at 2 m height, $D_1 = 25 \text{ kg m}^{-3}$, $D_2 = 250.0 \text{ kg m}^{-3}$ and $D_3 = 0.2 \text{ s m}^{-1}$. When $u_2 \leq 5 \text{ m s}^{-1}$, $\rho_{NS_w} = 0 \text{ kg m}^{-3}$.

$M$ is the snow melt mass flux that was computed with a temperature-index approach (Hock, 2003). Accordingly, $M = (I \cdot a)(T_A - T_\tau)\rho_S$.

$F$ is the melt freeze mass flux that was modelled with a coupled melt-freeze temperature-index approach. Accordingly, $F = (I^* \cdot e \cdot a)(T_A - T_\tau)\rho_W$.

The run-off $O$ was modelled with a matrix flow approach and it is equal to $O = \rho_W \alpha K_W$ where $\alpha = \alpha'(5.47 \cdot 10^5 \text{ m}^{-1}\text{ s}^{-1})$ (DeWalle and Rango, 2008) with $\alpha' = 3600 \text{ s h}^{-1}$. Following Colbeck (1972), $K_W$ was computed as $K_W = K S^{*3}$ in which $K$ is the intrinsic permeability of snow in $\text{m}^2$ and $S^*$ is the effective saturation degree of the mixture equal to $S^* = (S_r - S_{r_i})/(1 - S_{r_i})$ where $S_{r_i}$ is the irreducible saturation degree equal to $S_{r_i} = 0.02\rho_S/(\rho_W\phi)$ (Kelleners et al., 2009) and $S_r$ is the average saturation degree of the porous matrix equal to 1 when $h_W \geq \phi h_S$ while $S_r = h_W/(\phi h_S)$ otherwise (Avanzi et al., 2015). The intrinsic permeability is obtained using the parametrization proposed by Calonne et al. (2012) and it is equal to $K = 3R^2 \exp(-0.013\rho_S)$ in which $R$ is the equivalent sphere radius. The radius $R$ is defined as $R = 3/(SSA\rho_i)$ where $SSA$ is the specific surface area in $\text{m}^2 \text{ kg}^{-1}$ that was computed by Avanzi et al. (2015) adapting the formula proposed by Domine et al. (2007). Accordingly, $SSA = -30.82 \ln(\rho_S \cdot 10^{-3}) - 20.60$. When $S_r > 0.5$, to avoid numerical instability, the run-off was calculated with a kinematic wave approximation (De Michele et al., 2013) as $\rho_W \theta_W h_W^{1.25}$.

The firn melting $O_F$, that may occur only when the snowpack is absent, was modelled with a temperature-index approach and it is equal to $O_F = (I_F \cdot a)(T_A - T_\tau)\rho_F \delta(h_S)$.

$P_F$ is the effect of rain on firn that, when the snowpack is absent, causes an increase of $M_F$ when $T_A < 0 \,°C$ because rainfall is chilled to the firn temperature and a decrease when $T_A > 0 \,°C$ because the energy supplied by rain will be used to melt ice.





In the first case $P_F = \rho_W \cdot r$ while in the second case $P_F$ was set to zero due to its small contribution to mass balance (Doyle et al., 2015).

The terms $E_S, E_W, E_{MF}$ move the mass of the snowpack still on the ground at the end of each melt season inside the firn and they are equal to $E_j = \sum_i \rho_j \frac{h_j}{dt} \delta(t - t_i)$ with $j = S, MF, W$.

$Q$ is the mass of snow eroded by wind obtained from snow transport. The latter was computed adopting the parametrization proposed by Pomeroy et al. (1993) as the sum of a transport in saltation and a transport in suspension.

The saltation transport rate $Q_{salt}$ ($\mathrm{kg\,m^{-1}\,s^{-1}}$) occurs only when wind exceeds a given threshold and it is computed as follows:

$$Q_{salt} = \frac{0.68 \rho_a u_t^*}{u^* g}(u^{*2} - u_t^{*2}) \tag{A2}$$

where $\rho_a$ is the atmospheric density ($\mathrm{kg\,m^{-3}}$) and $u^*$ and $u_t^*$ are respectively the atmospheric friction velocity and the friction velocity applied to the snow surface at the transport threshold ($\mathrm{m\,s^{-1}}$). To move from the measured wind speed $u$ to $u^*$, knowledge of the aerodynamic roughness height $z_0$ is required. This passage is not straightforward since the value of $z_0$ during blowing snow events is different from the one during non transport conditions and it depends on friction velocity (Pomeroy and Gray, 1990). In order to avoid an iterative procedure, we adopted the approximation proposed by Pomeroy and Gray (1990). Accordingly, $u^* \approx 0.02264 u^{1.295}$ and $z_0 = \frac{0.1203 u^{*2}}{2g}$ where $u$ is 10 m wind speed ($\mathrm{m\,s^{-1}}$).

Suspension transport, that occurs only when particles are already in saltation, was computed as follows:

$$Q_{susp} = \frac{u^*}{\kappa} \int_{h^*}^{z_b} \eta(z) \ln\left(\frac{z}{z_0}\right) dz \tag{A3}$$

where $Q_{susp}$ is in $\mathrm{kg\,m^{-1}\,s^{-1}}$, $\kappa$ is the von Kármán constant equal to 0.4, $h^*$ is the lower boundary for suspension equal to $h^* = c_H u^{*1.27}$ (Pomeroy and Male, 1992) with $c_H = 0.08436\ \mathrm{m^{-0.27}\,s^{1.27}}$, $z_b$ is the top of the surface boundary-layer for suspended snow and $\eta(z)$ is the mass concentration of suspended snow ($\mathrm{kg\,m^{-3}}$) at height $z$. The mass concentration can be approximate as $\eta(z) = \eta(z_r) \exp(-A_Q((B_Q u^*)^{-0.544} - z^{-0.544}))$ (Pomeroy and Male, 1992) where $\eta(z_r)$ is the reference mass concentration for suspension set to 0.8 $\mathrm{kg\,m^{-3}}$ (Pomeroy and Male, 1992), $A_Q$ is equal to 1.55 $\mathrm{m^{0.544}}$ and $B_Q$ to 0.05628 $\mathrm{s^{-0.544}}$. $z_b$ was set to 5 m, since its value is typically between 5 m and 10 m (Déry and Taylor, 1996). The exact value is unimportant because of small mass fluxes at this height (Pomeroy et al., 1993). The snow erosion in the control volume of the model was set equal to the sum of these two transports.

The critical threshold, above which snow transport occurs, was computed adopting the formula proposed by He and Ohara (2017). Accordingly, the critical shear stress for snow movement can be computed as:

$$\tau_t = \frac{(8R \cdot C_g \cdot g)(\rho_S - \rho_a)\cos(\pi/3 - S) + (\pi C_c \varsigma)\left(\frac{C}{R^m} t_d\right)^{2/n}\left(\sin(\pi/3 + S) + \left(\frac{C}{R^m} t_d\right)^{1/n}\right)}{2\left(C_d \sin(\pi/3 - S) + C_l \cos(\pi/3 - S)\right)} \tag{A4}$$

where $R$ is the grain radius (m), $t_d$ is the time since deposition in seconds, $C_c, C_d, C_g$ and $C_l$ are dimensionless co-efficients set to 1, 4, $1.3\pi/6$ and 3.4 (He and Ohara, 2017), $\varsigma$ is the stress caused by cohesion of ice computed as $\varsigma = 1.51 \exp(0.44(T_A + 9)) + 6.8$ (Hosler et al., 1957) for temperatures between -20 °C and 0 °C with $\varsigma$ in $\mathrm{N\,m^{-2}}$ and $T_A$ in





°C and $S = \arcsin\left(\left(\frac{C}{R^m}t_d\right)^{1/n}\right)$. $C$, $m$ and $n$ are parameters that influence the rate of ice sintering, modelled following Maeno and Arakawa (2004). Accordingly, $C = C_0 \exp\left(\frac{-Q_s}{R_G(T_A+273.15)}\right)$ in which $R_G$ is the gas constant and $T_A$ is computed

as the average air temperature since deposition, $C_0 = 4.14 \cdot 10^{19} \ \mathrm{m}^3 \mathrm{s}^{-1}$ and $Q_s = 1.965 \cdot 10^5 \ \mathrm{J\,mol}^{-1}$. Finally, $m$ and $n$ are empirical parameters set to 2.9 and 5 respectively following the results of He and Ohara (2017). Once the critical shear stress is obtained it is possible to move to critical friction velocity as follows: $u_t^* = \sqrt{\tau_t/\rho_a}$. If wind speed is lower than the critical threshold, no erosion occurs and $Q$ is set to zero.

To implement the snow erosion routine, we proceeded as explained in the following. When the first solid precipitation event

occurs in a time step, the amount of new snow on the ground at the end of the time step, $S_A$ ($\mathrm{kg\,m}^{-2}\mathrm{h}^{-1}$), is saved along with the time of deposition and $\rho_{NS}$ of the event. During the following step four different situations are possible: (1) a new snow event occurs in the time step. In this case $S_A$ is moved into a vector $\boldsymbol{S_R}$ with its time of deposition and $\rho_{NS}$. $S_A$ is recomputed as $\rho_{NS} \cdot s$; (2) $T_A > 0$ °C. In this case $S_A$ and $Q$ are set to 0 $\mathrm{kg\,m}^{-2}\mathrm{h}^{-1}$ and all the old snow events memorized in $\boldsymbol{S_R}$ are removed; (3) $T_A < 0$ °C and $Q < S_A$. In this case $S_A$ is set to $S_A = S_A - Q$; (4) $T_A < 0$ °C and $Q > S_A$. In this case, if $\boldsymbol{S_R}$

has no elements, $Q$ is set equal to $S_A$ and $S_A$ to 0 $\mathrm{kg\,m}^{-2}\mathrm{h}^{-1}$, otherwise the difference between $Q$ and $S_A$ is subtracted from the most recent event in $\boldsymbol{S_R}$, given that wind speed is higher than the threshold recomputed with the characteristics of that event, and this event is removed from $\boldsymbol{S_R}$. This is repeated until an event in $\boldsymbol{S_R}$ that cannot be eroded by wind is encountered or the total amount of snow eroded in that time step reaches $Q$. In the latter case the actual transport is $Q$ while in the former $Q$ is given by the total amount of snow eroded before reaching the non erodible layer. The new $S_A$ is the amount of snow

associated with the last event considered.

Given that $M_j = \rho_j h_j$ and $\rho_k = \mathrm{const}$ with $j = S, MF, W$ and $k = MF, W$, after some algebra we can move from Eqs. A1a–A1d to Eqs. 2a–2d.

## A2 Snow densification

The densification of dry snow due to compaction was modelled adopting the formula proposed by Liston et al. (2007). Accord-

600 ingly,

$$\frac{d\rho_S}{dt} = (c \cdot A_1 \cdot U)\rho_S \exp(-B \cdot (T_\tau - T_S) - A_2 \cdot \rho_S) \tag{A5}$$

where $c = 0.10 \cdot 3600 \ \mathrm{s\,h}^{-1}$, $A_1 = 0.0013 \ \mathrm{m}^{-1}$, $A_2 = 0.021 \ \mathrm{m}^3\mathrm{kg}^{-1}$, $B = 0.08 \ \mathrm{K}^{-1}$ and $U$ is the wind speed contribution ($\mathrm{m\,s}^{-1}$). For wind speeds $\geq 5 \ \mathrm{m\,s}^{-1}$, $U = E_1 + E_2(1.0 - \exp(-E_3(u_2 - 5.0)))$ with $E_1$, $E_2$ and $E_3$ equal to 5.0 $\mathrm{m\,s}^{-1}$, 15.0 $\mathrm{m\,s}^{-1}$ and 0.2 $\mathrm{s\,m}^{-1}$, respectively, and $u_2$ the wind speed at 2 m height. For wind speed $< 5 \ \mathrm{m\,s}^{-1}$, $U = 1 \ \mathrm{m\,s}^{-1}$. Adding

the densification due to mass variation (see De Michele et al. (2013)) the total densification rate can be computed as follows:

$$\frac{d\rho_S}{dt} = (c \cdot A_1 \cdot U)\rho_S \exp(-B \cdot (T_\tau - T_S) - A_2 \cdot \rho_S) + \frac{\rho_{NS} - \rho_S}{h_S}s \tag{A6}$$

where we assumed that melting events and snow erosion occur at $\rho_S = \mathrm{const}$.





## A3 Arnaud et al. (2000) model

The model of AR separates the densification of firn into three stages. The first stage is governed by settling and it is modelled
by Bréant et al. (2017) adapting the equation proposed by Alley (1987). The second stage, that starts when the relative density
$D = \rho_F / \rho_i$ equals $D_0$, is dominated by power law creep and it is modelled following Arzt (1982) and Arzt et al. (1983). Grains
are considered as spheres and each sphere is allowed to increase in radius around fixed centres. Starting form an initial radius
$l$, the new radius $l'$ (in units of the initial particle radius $l$) is $l'(D) = (D/D_0)^{1/3}$. The growth of spheres increases the number
of particle contacts $Z$ from the initial value $Z_0$ to $Z(D) = Z_0 + b(l' - 1)$ in which $b = 15.5$. The overlap due to the growth of
particles produces an excess volume of material. This excess is distributed uniformly around the portion of the surface of the
spheres not in contact. From this excess volume, it is possible to calculate the new radius $l''$ as

$$l'' = l' + \frac{4Z_0(l'-1)^2(2l'+1) + b(l'-1)^3(3l'+1)}{12l'\left(4l' - 2Z_0(l'-1) - b(l'-1)^2\right)} \tag{A7}$$

The average contact area (in unit of $l^2$) can be obtained averaging over all of existing contacts:

$$a(D) = a(l'') = \frac{\pi}{3Zl'^2}\left(3(l''^2 - 1)Z_0 + l''^2 b(2l'' - 3) + b\right) \tag{A8}$$

The value of $Z_0$ for a given value of $D_0$ is obtained, as proposed by Arnaud et al. (2000), assuming that the effective stress
$P^* = (4\pi P)/(a_c Z D)$ approaches $P$ as $D$ tends to 1. The third stage begins when pores start becoming isolated $(D > D_c)$
and densification is calculated considering the deformation of ice shells surrounding cylindrical pores (Wilkinson and Ashby,
1975). As for Eq. 2e, the total densification rate is obtained adding the densification due to new mass addition (Eq. 2f).

## Appendix B: List of all the symbols used





**Table B1.** List of the symbols used in the mass fluxes of the snow-firn model with the exclusion of snow erosion (from A to S).

| Symbol | Description | Type | Unit |
|--------|-------------|------|------|
| $a$ | degree hour parameter | **calibration parameter** | $\mathrm{m\,h^{-1}\,^{\circ}C^{-1}}$ |
| $D_1, D_2$ | constants governing the influence of wind in fresh snow density | constant | $\mathrm{kg\,m^{-3}}$ |
| $D_3$ | constants governing the influence of wind in fresh snow density | constant | $\mathrm{m^{-1}\,s}$ |
| $E_{MF}, E_S, E_W$ | mass flux due to the transformation of snow in firn | variable | $\mathrm{kg\,m^{-2}\,h^{-1}}$ |
| $e$ | melt-freeze factor | **calibration parameter** | $-$ |
| $F$ | melt freeze mass flux | variable | $\mathrm{kg\,m^{-2}\,h^{-1}}$ |
| $h$ | snowpack height | variable | m |
| $h_F$ | firn height | variable | m |
| $h_{MF}$ | height of ice inside the snowpack | variable | m |
| $h_S$ | dry snow height | variable | m |
| $h_W$ | height of water inside the snowpack | variable | m |
| $I, I^*, I_F$ | multiplicative function | function | |
| $K$ | intrinsic permeability of snow | variable | $\mathrm{m^2}$ |
| $K_W$ | intrinsic permeability of water in snow | variable | $\mathrm{m^2}$ |
| $k$ | constant | constant | m |
| $M$ | snow melt mass flux | variable | $\mathrm{kg\,m^{-2}\,h^{-1}}$ |
| $M_F$ | firn mass | variable | $\mathrm{kg\,m^{-2}}$ |
| $M_{MF}$ | mass of ice inside the snowpack | variable | $\mathrm{kg\,m^{-2}}$ |
| $M_S$ | dry snow mass | variable | $\mathrm{kg\,m^{-2}}$ |
| $M_W$ | mass of water inside the snowpack | variable | $\mathrm{kg\,m^{-2}}$ |
| $O$ | run-off rate | variable | $\mathrm{kg\,m^{-2}\,h^{-1}}$ |
| $O_F$ | firn melt mass flux | variable | $\mathrm{kg\,m^{-2}\,h^{-1}}$ |
| $P_F$ | variation of mass due to rain on firn | variable | $\mathrm{kg\,m^{-2}\,h^{-1}}$ |
| $P_R$ | mass flux of liquid precipitation | variable | $\mathrm{kg\,m^{-2}\,h^{-1}}$ |
| $P_S$ | mass flux of solid precipitation | variable | $\mathrm{kg\,m^{-2}\,h^{-1}}$ |
| $R$ | grain radius | variable | m |
| $r$ | liquid precipitation rate | variable | $\mathrm{m\,h^{-1}}$ |
| $S^*$ | effective saturation degree of the snowpack | variable | $-$ |
| $S_r$ | average saturation degree of the porous matrix | variable | $-$ |
| $S_{r_i}$ | irreducible saturation degree | variable | $-$ |
| $s$ | solid precipitation rate | variable | $\mathrm{m\,h^{-1}}$ |
| $SSA$ | specific surface area | variable | $\mathrm{m^2\,kg^{-1}}$ |
| $SWE$ | snow water equivalent | variable | m |





**Table B2.** List of the symbols used in the mass fluxes of the snow-firn model with the exclusion of snow erosion (from T to Z and Greek letters).

| Symbol | Description | Type | Unit |
|--------|-------------|------|------|
| $T_A$ | air temperature | variable | $^\circ$C |
| $T_\tau$ | threshold temperature for melting | here treated as constant | $^\circ$C |
| $u_2$ | 2 m wind speed | variable | $\mathrm{m\,s^{-1}}$ |
| $V_F$ | firn volume | variable | $\mathrm{m^3}$ |
| $V_{MF}$ | volume of ice inside the snowpack | variable | $\mathrm{m^3}$ |
| $V_S$ | dry snow volume | variable | $\mathrm{m^3}$ |
| $V_W$ | volume of water inside the snowpack | variable | $\mathrm{m^3}$ |
| $\alpha$ | constant governing run-off rate | constant | $\mathrm{m^{-1}\,s^{-1}}$ |
| $\alpha'$ | time conversion constant | constant | $\mathrm{s\,h^{-1}}$ |
| $\rho$ | bulk density of snow | variable | $\mathrm{kg\,m^{-3}}$ |
| $\rho_F$ | firn density | variable | $\mathrm{kg\,m^{-3}}$ |
| $\rho_i$ | ice density | here treated as constant | $\mathrm{kg\,m^{-3}}$ |
| $\rho_{NS}$ | fresh snow density | variable | $\mathrm{kg\,m^{-3}}$ |
| $\rho_{NS_0}$ | fresh snow density without wind | variable | $\mathrm{kg\,m^{-3}}$ |
| $\rho_{NS_w}$ | fresh snow density increase due to wind | variable | $\mathrm{kg\,m^{-3}}$ |
| $\rho_S$ | dry snow density | variable | $\mathrm{kg\,m^{-3}}$ |
| $\rho_W$ | water density | here treated as constant | $\mathrm{kg\,m^{-3}}$ |
| $\theta_W$ | volumetric liquid water content | variable | – |
| $\phi$ | porosity | variable | – |





**Table B3.** List of the symbols used to compute snow erosion.

| Symbol | Description | Type | Unit |
|---|---|---|---|
| $A_Q$ | constant governing the mass concentration of suspended snow | constant | $\mathrm{m}^{0.544}$ |
| $B_Q$ | constant governing the mass concentration of suspended snow | constant | $\mathrm{s}^{-0.544}$ |
| $C$ | constant governing ice sintering function of $T_A$ | constant | $\mathrm{m}^3\,\mathrm{s}^{-1}$ |
| $C_0$ | constant governing ice sintering | constant | $\mathrm{m}^3\,\mathrm{s}^{-1}$ |
| $C_c, C_d, C_g, C_l$ | coefficient governing cohesive force, drag, form and lift coefficient | parameter | $-$ |
| $c_H$ | coefficient influencing the lower boundary height for suspension | constant | $\mathrm{m}^{-0.27}\,\mathrm{s}^{1.27}$ |
| $g$ | gravitational acceleration | constant | $\mathrm{m}\,\mathrm{s}^{-2}$ |
| $h^*$ | lower boundary for suspension | variable | m |
| $m$ | parameter governing ice sintering | parameter | $-$ |
| $n$ | parameter governing ice sintering | parameter | $-$ |
| $Q$ | snow erosion | variable | $\mathrm{kg}\,\mathrm{m}^{-2}\,\mathrm{h}^{-1}$ |
| $Q_s$ | activation energy | constant | $\mathrm{J}\,\mathrm{mol}^{-1}$ |
| $Q_{salt}$ | snow transport in saltation | variable | $\mathrm{kg}\,\mathrm{m}^{-1}\,\mathrm{s}^{-1}$ |
| $Q_{susp}$ | snow transport in saltation | variable | $\mathrm{kg}\,\mathrm{m}^{-1}\,\mathrm{s}^{-1}$ |
| $R$ | grain radius | variable | m |
| $R_G$ | gas constant | constant | $\mathrm{J}\,\mathrm{K}^{-1}\,\mathrm{mol}^{-1}$ |
| $S_A$ | mass of the most recent non eroded snow events | variable | $\mathrm{kg}\,\mathrm{m}^{-2}\,\mathrm{h}^{-1}$ |
| $\boldsymbol{S_R}$ | vector of non eroded snow events | variable | $\mathrm{kg}\,\mathrm{m}^{-2}\,\mathrm{h}^{-1}$ |
| $t_d$ | time since deposition | variable | s |
| $u$ | 10 m wind speed | variable | $\mathrm{m}\,\mathrm{s}^{-1}$ |
| $u^*$ | atmospheric friction velocity | variable | $\mathrm{m}\,\mathrm{s}^{-1}$ |
| $u_t^*$ | critical friction velocity | variable | $\mathrm{m}\,\mathrm{s}^{-1}$ |
| $z$ | altitude | variable | m |
| $z_0$ | aerodynamic roughness length | variable | m |
| $z_b$ | top boundary for suspension | here treated as constant | m |
| $z_r$ | reference height for mass concentration of suspended snow | constant | m |
| $\eta$ | mass concentration of suspended snow | variable | $\mathrm{kg}\,\mathrm{m}^{-3}$ |
| $\kappa$ | von Kármán constant | constant | $-$ |
| $\rho_a$ | atmospheric density | here treated as constant | $\mathrm{kg}\,\mathrm{m}^{-3}$ |
| $\rho_S$ | dry snow density | variable | $\mathrm{kg}\,\mathrm{m}^{-3}$ |
| $\varsigma$ | stress due to ice cohesion | variable | Pa |
| $\tau_t$ | critical shear stress for erosion | variable | Pa |





**Table B4.** List of the symbols used in AR.

| Symbol | Description | Type | Unit |
|--------|-------------|------|------|
| $A$ | creep constant function of $T_F$ | constant | $\mathrm{Pa^{-3}\,h^{-1}}$ |
| $A_0$ | constant governing firn densification | constant | $\mathrm{Pa^{-3}\,h^{-1}}$ |
| $a_c$ | average contact area | variable | $-$ |
| $b$ | parameter in firn densification | parameter | $-$ |
| $D$ | relative firn density | variable | $-$ |
| $D_0$ | relative density between first and second stage of firn densification | here treated as constant | $-$ |
| $D_c$ | close-off density | here treated as constant | $-$ |
| $D_D$ | relative surface density | variable | $-$ |
| $l', l''$ | firn grain radius in units of the initial radius $l$ | variable | $-$ |
| $P$ | overburden pressure | variable | Pa |
| $P^*$ | effective stress | variable | Pa |
| $P_b$ | pressure in the bubbles | variable | Pa |
| $P_c$ | atmospheric pressure at the close-off | here treated as constant | Pa |
| $Q_1, Q_2$ | activation energy | constant | $\mathrm{J\,mol^{-1}}$ |
| $R_G$ | gas constant | constant | $\mathrm{J\,K^{-1}\,mol^{-1}}$ |
| $T_F$ | average firn temperature | variable | $^\circ$C |
| $Z$ | number of particle contacts | variable | $-$ |
| $Z_0$ | initial number of particle contacts | constant | $-$ |
| $\gamma$ | parameter function of $\gamma', T_F$ | parameter | $\mathrm{Pa^{-1}\,h^{-1}}$ |
| $\gamma'$ | parameter governing firn densification | **calibration parameter** | $\mathrm{Pa^{-1}\,h^{-1}}$ |
| $\rho_F$ | firn density | variable | $\mathrm{kg\,m^{-3}}$ |
| $\rho_i$ | ice density | here treated as constant | $\mathrm{kg\,m^{-3}}$ |





**Table B5.** List of the symbols used in HL.

| Symbol | Description | Type | Unit |
|---|---|---|---|
| $k_0$ | constant governing firn densification | constant | $\mathrm{m}^{-1}$ |
| $k_1$ | constant governing firn densification | constant | $\mathrm{m}^{-0.5}\,\mathrm{y}^{-0.5}$ |
| $R_G$ | gas constant | constant | $\mathrm{J\,K^{-1}\,mol^{-1}}$ |
| $T_F$ | average firn temperature | variable | $^{\circ}\mathrm{C}$ |
| $\rho_D$ | surface density | variable | $\mathrm{kg\,m^{-3}}$ |
| $\rho_F$ | firn density | variable | $\mathrm{kg\,m^{-3}}$ |
| $\rho_i$ | ice density | here treated as constant | $\mathrm{kg\,m^{-3}}$ |
| $\omega$ | mean annual accumulation | variable | $\mathrm{kg\,m^{-2}\,y^{-1}}$ |

**Table B6.** List of the symbols related to snow densification rate.

| Symbol | Description | Type | Unit |
|---|---|---|---|
| $A_1$ | constant governing snow densification | constant | $\mathrm{m}^{-1}$ |
| $A_2$ | constant governing snow densification | constant | $\mathrm{m}^{3}\,\mathrm{kg}^{-1}$ |
| $B$ | constant governing snow densification | constant | $\mathrm{K}^{-1}$ |
| $c$ | constant governing snow densification | constant | $\mathrm{s\,h^{-1}}$ |
| $E_1, E_2$ | constants governing the influence of wind in snow densification | constant | $\mathrm{m\,s^{-1}}$ |
| $E_3$ | constants governing the influence of wind in snow densification | constant | $\mathrm{m^{-1}\,s}$ |
| $T_S$ | average snow temperature | variable | $^{\circ}\mathrm{C}$ |
| $T_\tau$ | threshold temperature for melting | here treated as constant | $^{\circ}\mathrm{C}$ |
| $U$ | wind speed contribution to snow densification | variable | $\mathrm{m\,s^{-1}}$ |
| $u_2$ | 2 m wind speed | variable | $\mathrm{m\,s^{-1}}$ |
| $\rho_S$ | dry snow density | variable | $\mathrm{kg\,m^{-3}}$ |





**Table B7.** List of other symbols used throughout the paper.

| Symbol | Description | Type | Unit |
|--------|-------------|------|------|
| $H$ | mountain height | constant | m |
| $MAAT$ | mean annual air temperature | constant | °C |
| $MAFT$ | mean annual firn temperature | constant | °C |
| $p$ | daily precipitation | variable | $\mathrm{mm\,d^{-1}}$ |
| $SD$ | observed snow depth | variable | m |
| $T_0$ | surface temperature | variable | °C |
| $T_A$ | air temperature | variable | °C |
| $t_i$ | time instant at the end of hydrological year $i$ | constant | h |
| $u$ | 10 m wind speed | variable | $\mathrm{m\,s^{-1}}$ |
| $z_M$ | maximum firn depth influenced by air temperature | here treated as constant | m |
| $z_m$ | altitude of maximum precipitation | here treated as constant | m |
| $\Delta t$ | time step | constant | h |
| $\delta$ | Dirac delta function | function | |



*Author contributions.* CDM and FB conceived the model. FB took care of data, and developed the case study. FB wrote a first draft of the manuscript. FB and CDM reviewed the manuscript.

*Competing interests.* No competing interests are present.

*Acknowledgements.* We would like to thank ARPA Piemonte for the meteorological data used in this study, with particular thanks to Manuela Bassi. Gratitude is also due to Carlo Licciulli and Josef Lier for ice core data and to Pascal Bohleber for the information about Colle Gnifetti. One last thank you to Scenari Digitali and Rifugi Monterosa for additional information about Capanna Regina Margherita.





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
