# Peer review of "A local model of snow-firn dynamics and application to Colle Gnifetti site"

_The Cryosphere, 2021_

## Referee Comment (RC2)

**General comments**

The paper 'A local model of snow-firn dynamics and application to Colle Gnifetti site' by Fabiola Banfi and Carlo De Michele is a study based on the development of a snow/firn numerical model to investigate the evolution of density and snow accumulation at Colle Gnifetti. The model essentially relies on mass conservation of the various components of the system (i.e. dry snow, liquid water, firn), but it includes several fully empirical or semi-empirical laws to simulate processes such as wind erosion, firn densification, melting, snowpack densification due to wind... On the other hand, several important processes for snowpack evolution are not taken into account: wind re-deposition, water percolation, heat transport, radiative transfer, snow metamorphism... The flow of firn/ice is also not included. There are two versions of the model: one considering the firn column as a single layer, and for which the output in terms of density is an averaged value (which rises questions, see below); another one in which the firn is decomposed in multiple layers which enables to get a vertical density profile. Two models of firn densification are also tested. The model is forced with meteorological data (air temperature, wind speed, precipitation) recorded at near-by weather stations. The data recorded at one of the considered weather station are used to calibrate some of the model parameters. The two versions of the model, with the two approaches for firn densification, are then tested at Colle Gnifetti. Model outputs are compared to observations of density profile and snow accumulation from cores drilled on site. The modelled order of magnitude are correct but it is difficult to draw firm conclusions regarding the capabilities of the model due to high spatio-temporal variability of density profiles and accumulation observed at the site. One interresting result of the study is that, counter-intuitively, the positive contribution to surface mass balance at Colle Gnifetti occurs in summer, as all winter snow is blown away by wind.

As stated in the introduction, including the contribution of glacier run-off when projecting the future evolution of hydrological regimes in mountainous regions does indeed sound to be of prime importance in the context of global warming. Although I am not very familiar with hydrological models, I can easily imagine that this requires the development of simplified models of snow/firn able to simulate the amount of Snow Water Equivalent (SWE) stored in the system and its evolution over time, disregarding all the complexity associated to the most detailed models such as SNOWPACK or Crocus. This makes this study relevant and the paper probably ought to be published. This being said, I have a fair amount of major criticisms to formulate, both on the methodology and on the conclusions that are drawn from the numerical experiments.

First of all, I think that the model is much more empirical than what is claimed in the paper. Of course the core of the model relies on mass conservation, but most of the sink/source contributions are based on empirical approaches. This clearly appears when reading appendix A: melting is based on a temperature-index approach and does not care about the actual distribution of enthalpy within the sytem; the run-off relies on a matrix flow approach; the wind-drift is also based on laws relying on several parameterizations; on the contrary to what is written in the paper, I don't think that snow densification is modelled through the iterative resolution of mass and momentum conservation (i.e. $\partial \rho / \partial t + \text{div}(\rho v) = 0$ and $\text{div}\,\sigma + \rho g = 0$), but via a semi-empirical formula; same goes for firn densification (see next paragraph)... This results in a large amount of parameters, whose values have to be set. You claim that only two parameters, $a$ and $e$, require calibration as all other values can be found in literature. I found this argument rather poor as, most of the time, values proposed in the literature are poorly constrained and are usually valable only in a very specific context which does not necessaily correspond to yours. In a way, this point is somewhat addressed for what regards the firn densification models that were derived from polar cores when you say that they can be apply in your context due to low firn temperature and low accumulation, mimicking polar conditions (l. 176). I am not saying that these empirical approaches make the study worthless, but I would assume more clearly this state of fact in the manuscript and avoid arguing that it "has the advantage not only that the model is less site specific but also that it can be applied to sites with a small amount of data, like most of the high-altitude sites". Regarding the calibration of parameter $a$, I find the description of the calibration procedure a bit confused (see specific comments). Also, you claim that "the parameter $\gamma'$ that governs firn densification rate in AR, instead, does not need calibration since it is fixed once the mean accumulation and MAFT are set". But then, you explain that the value of this parameter was fixed by running AR in steady-state condition such that there is no discontinuity of the densification rate between the first and second stage of densification. But this is a form of calibration, right ?

My second main point regards the firn densification. I really have a hard time to understand how the density of firn and its evolution is actually computed. If I understood well, you get a single value of density per layer of firn in the multi-layer version and one single value for the whole firn column in the one-layer version ? But, in the AR model, density is an explicit function of overburden pressure, which itself should be a continuous function of depth.

Therefore, you should get a continuous profile of density with depth, which seems to be the case when looking at Fig. 4. I can somehow understand that, in the multi-layer case, the density of a given layer is obtained considering the weight of overlying layers. But how do you get a value for the single layer case ? How is overburden stress defined in that case ? If it is an averaged value, then how is the firn column over which the average is made defined ? Is it the whole column from firn/snowpack interface down to the bedrock ? I think this needs to be clarified in Section 2.4. At the very least, you should state explicitly how overburden pressure is calculated.

One of the consequence of my lack of understanding of how firn densification is actually computed is that I do not understand any of the Figures where density is plotted as a function of the layer deposition year (i.e. Figs 8, 9, 11). I can understand a plot of density versus depth (as in Fig. 4), but the plot of density versus year of deposition does not really make sense to me. Indeed, if one considers the density of the layer corresponding to snow deposited on a given year, this density will keep increasing over time due to the ever-increasing weight supported by this layer.Thus, the density of layer deposited, say, in 2005 will be smaller if assessed in 2015 than if assessed in 2020. There is an attempt of explanation of this point in l.355 to 359, but it is not very clear to me. I would reformulate to make it much clearer.

Regarding the conclusions of your study, I think that your main result, i.e. that the snow that is preserved and contribute positively to mass balance is the snow that fall in summer months as winter snow is eroded by wind, is an interesting result, but most likely very site-specific and cannot be extended to the scale of a whole massif or mountain range. Therefore, I think that the sentence (l. 421 to 424) "This, in fact, could lead, as already suggested by Alean et al. (1983), to a counterintuitive response in a scenario of higher temperatures, since the increased melting and the reduced fraction of solid events with respect to total precipitation will be accompanied by a reduced snow erosion. An increase of the positive net balances may in turn influence ice avalanche activity" should be reformulated to make it clear that this is specific to Colle Gnifetti. In addition, I don't really see the link with ice avalanche activity.

Furthermore, I find some of your statements in the conclusions Section a bit too strong. I think this Section requires a fair amount of rewriting.

On the structure of the paper, I think it is generally well-structured apart for Section 3.4 which gathers information regarding the calibration of parameters $a$, $e$ (which is not calibrated in the end), and $\gamma'$ (which is allegedly not calibrated but actually is), as well as values of model parameters that are fixed based on the literature. My opinion is that the latter should rather go in the corresponding subsections of Section 2 (especially Section 2.4) where most of the chosen values for the other model parameters are already given. I would keep section 3.4 to describe the calibration procedure of $a$ and $\gamma'$ only. And I would rename it as "Calibration of model parameters", or something like that.

Finally, regarding the form of the paper, although I am not a native English speaker myself, I think that the paper is rather well-written, although some sentences sound a bit odd to me (see specific comments). For example, the way you use "Accordingly," (lots of occurrences in Appendix A among other) to specify that you are following the approach of a previously mentioned reference does not sound natural to me. But I might be wrong on that point. Also, I think that a lot of commas are missing. I have noted down some of them in the specific comments but it is far from being exhaustive.

**Specific comments**

P1 L8: "We observed" $\rightarrow$ I would reformulate to make clear that this is a modelling result and not an observation.

P1 L18: polar regions.

P1 L23-25: This sentence is a bit complicated. I would reformulate.

P2 L26: glacier melt

P2 L29: "and release with implications" $\rightarrow$ and release, with implications

P2 L31: assess

P2 L40: "both components also in basins that in the present are highly glacierized." → both components, even in basins that are currently highly glacierized.

P2 L45: "In addition they are" → In addition, they are

P2 L47: "to reproduce the important processes" → to reproduce important processes (?)

P2 L47: "In the present work we give" → In the present work, we give

P2 L50: "momentum balance and rheological equations" → I don't really see any equations related to the momentum conservation ($\mathrm{div}\boldsymbol{\sigma} + \rho g = 0$) or firn rheology ($\mathbf{S} = \boldsymbol{\sigma} + p\mathbf{I} = 2\eta\dot{\boldsymbol{\epsilon}}$)

P2 L50: "and they estimate the snowpack and firn characteristics" → and they govern the evolution of the snowpack and firn characteristics

P2 L55: "it provides more insight on" → it captures better (?)

P2 L59: "In order to test the model a high" → In order to test the model, a high

P3 L62: "in the following way" → as follows

P3 L62: "we provide" → we present *or* we describe

P3 L63: "and discussion" → and discuss them

P3 L84: "and metamorphosis of grains not driven by wind" → I don't think there is any representation of snow metamorphism in your model

P4 L100-101: "the version presented in this work includes the effect of wind both on mass balance and densification" → For mass balance, it is only as a sink term (erosion) as wind re-deposition is not represented

P4 L105: properties

P4 L113: Point (2) is not very clear, could you precise ?

P5 L122: I would stress the fact that multi-layer refers to multi-layer of firn and that the snowpack is always treated as a single layer (on the contrary to more detailed snow models such as SNOWPACK or Crocus).

P5 L119: "In both cases we neglected" → In both cases, we neglected

P5 L119: "In order to separate snow from firn we refer" → In order to separate snow from firn, we refer

P6 Eq (2b): I don't really understand how the last term of the right-hand side of this equation is treated as the firn is said to be impermeable. How do you transfer the remaining liquid water to the firn then ?

P7 L170: "HL was optimized" → HL model was derived from cores...

P7 L171: "AL" → AR

P7 L171: "AR was optimized" → AR model was derived from cores...

P7 L176-177: I would reformulate this sentence. What you mean is that the low accumulation and low MAFT observed at CG justifies the use of the AR and HL models as such, despite the fact that they were derived for polar conditions while CG is an alpine site.

P8 Eq (4) Top rhs member: Remove the Pa (if that stands for Pascal, pressures should naturally be given in Pascals) and check wether the $10^4$ has to stay or not

P8 Eq (4): In the text you are mentionning three densification regimes but there are four equations here. Could you clarified that point ?

P8 L184: "The overburden P" $\rightarrow$ The overburden pressure P

P8 L186: "(see Appendix A for the expression of $a_c$ and $Z$)" $\rightarrow$ You are referring to Appendix A3 (be more specific about the appadendix, here and elsewhere in the text). Also $a_c$ is denoted $a$ in Appendix A3, which is not consistent.

P8 L196: It is not clear why the information regarding the transition density for HL is given here while the same information for AR is given at the end of Section 3.4. I would bring the latter back here (see Major Comments).

P8 L199: Idem, I would put the information regarding the value of $\rho_D$ here instead of Section 3.4

P8 L201: "In this way the passage from snow to firn densification is driven by snow characteristics rather than snow age" $\rightarrow$ 1/ In this way, ... 2/ I don't really see where snow age appears as the driving factor for densification from Eq. (4) and (5).

P8 L202: "corresponds to the average surface density" $\rightarrow$ Averaged over what ? Time ? Observations from cores ?

P9 L207: "At depths higher than $z_M$" $\rightarrow$ This formulation is confusing. I suggest 'deeper than $z_M$' or 'below $z_M$'

P10 Fig. 2: The map is a bit difficult to read. At least, I would remove the dotted lines (what is it ?) and the line made of crosses (boundary between Swiss and Italy ?) or add a legend to tell what they mean.

P11 Fig. 3: It would be great to have a box on the background map to know exactly to what region the zoom-in of the lower right corner corresponds. Also, the fact that it is a zoom-in should be precised in the caption. You can also mention that CM appears on Fig. 2 as well, which will help the reader to localize the study area on this map.

P11 L260-261: "datum" $\rightarrow$ Although, it might be correct from a grammatical point of view, this formulation sounds quite odd to me. I would suggest data or information

P13 L283: "smoothing the snow depth with a moving average whose window size was calibrated" $\rightarrow$ Not very clear, could you clarified ?

P14 Section 3.4: See Major Comments

P14 L295: "The parameter $a$" $\rightarrow$ I would remind the physical meaning of this parameter

P14 L296-301: The procedure you are following to calibrate $a$ is not very clear. First, I don't understand why the wind contribution is not accounted for as it should explain part of the ablation, the other part being due to melting. Then, if I understand well, you are calculating one optimal value of $a$ per hydrological year, and then, for each thus obtained value of $a$, you compare observed snow depth to snow depth modelled for all the hydrological years except the one from which the considered value of $a$ has been derived. Is that correct ? If yes, I think that this procedure makes sense, but what I find surprising is that you don't use these comparisons between model outputs and observations to set the final value of $a$ but you simply take the median of all the $a$ you have calculated. Why is that ?

P14 L304-306:"Its value is in fact chosen in order to have a continuous densification ratebbetween first and second stage of densification. For each of the available ice cores, with the exception of CG11, we computed the parameter $\gamma'$ running AR in a steady-state condition (Bader, 1954) using the mean accumulation reported in Table 2." $\rightarrow$ In other words, you are calibrating $\gamma'$, right ?

P14 L308-312: The way you are setting the value of $\rho_D$ for the various cores sounds a bit random as presented here.

P14 L319: The parameterization of SSA given here is not consistent with the one given in Appendix A1.

P15 L323: Why does the unit of NSE, RMSE and MBE is different from the original unit of $a$ ?

P15 L324: What is the SNOTEL network?

P15 L329-331: I have the feeling that a re-calibration of parameter $a$ would be required if the value of $e$ is changed to be consistent. Is it not the case ?

P15 L345: "are provided in Fig. 6" $\rightarrow$ You should state or remind the time period of simulation from which Fig. 6 is obtained.

P17 L351: I am not sure that the median of wind speed gives a good picture of how windy a month was. But I have nothing else to propose.

P17 Section 4.4.1: As explained in the Major Comments, I think that this Section needs a fair amount of rewriting and, therefore, I do not make specific comments here.

P18 Section 4.4.2: Same as above.

P18 Fig. 6: For the lower right Figure, I think what is represented is 100 $\times$ (Solid precip. - Eroded snow)/ Solid precip., am I correct ? If yes, you could write it explicitly.

P19 Fig. 7: I am not sure this Fig. brings relevant additional information.

P20 L386: Why comparing HL and AR between each other and not to observations ? In addition, in its current state, this sentence seems to countradict the one just before.

P21 L395: "to fully represent the site, besides, ice core..." $\rightarrow$ to fully represent the site. In addition, ice core...

P21 L408-409: How do we compare cm to $\mathrm{kg\,m^{-2}\,yr^{-1}}$ ? More broadly speaking, a choice has been made to give all results in terms of accumulation in $\mathrm{kg\,m^{-2}\,yr^{-1}}$, but I think that m SWE is often easier to apprehend. But that's only my opinion.

P21 L415:"On the contrary, it is less acknowledged in other fields" $\rightarrow$ I would remove this sentence.

P21-22 L.421-424: See Major Comments.

P22 Fig. 10:The choice of the colors is not ideal.

P23 L442:"The annual behaviour" $\rightarrow$ You mean the inter-annual variability, don't you ?

P23 L451:"that causes the two..." $\rightarrow$ that are very likely responsible for the significantly different behaviors of the two ice cores...

P23 L456-457:"along with the snow redistribution due to gravity" $\rightarrow$ What do you mean ?

P23 L460-461:"We can instead see an improvement in the ability of the model to follow year-to-year variations in the period 2008-2014" $\rightarrow$ It is not very obvious.

P24 L473:"optimized" $\rightarrow$ derived from (?)

P24 L474:"Results in Fig. 9, on the contrary, show a better and more robust performance of the model with" $\rightarrow$ one cannot say that it is robust when it changes depending on the core being considered.

P24 L485:"to asses the influence of meteorological variables on snow and firn characteristics." → (1) to assess; (2) How this could be helpful ?

P25 L486:"therefore moving the boundary of the model from surface accumulation and density to hourly meteorological series" → This is an interresting way of formulating what you have been doing in this paper. I think a similar formulation of the problematic you are addressing would be welcomed in the introduction as well.

P25 L489:"Both the two model's versions have a parsimonious parametrization, with only two parameters (a, e) to be estimated." → I find this sentence largely overstated. Your model (both versions) largely relies on parameterization, see my Major Comments.

P25 L490-491:"All the other parameters were, instead, taken from literature" → Maybe, but most of them are far from being well-constrained !

P25 L491-493:I find this argumentation a bit awkward. It is like if you were arguing that it is better to have a model that performs badly whatever the conditions than to have a model that has good performances only when used in its domain of validity. It is not very convincing.

P25 L495:"we presented a general modelling that includes all the selected processes" → That is a tautology, isn't it ?

P25 L506:"and the spatial variation of solar radiation" → Not only the spatial variation but solar radiation at all.

---

## Author Comment (AC1)

Dear Referee #1,

thank you for your comments.

We will first address the major comment regarding the position of the single- and multi-layer version of the model and then we will report the reply to the minor comments.

The aim of the model is to integrate the different processes involved in the transformation of snow into firn. The choice to present both model versions in the new manuscript was driven by the consideration that the model is intended to be used in two different contexts: (1) with the multi-layer version it is possible to assess the evolution of firn under climate changes, since the model uses as input meteorological forcings (i.e., precipitation, temperature and wind speed) rather than snow accumulation and density. In this sense, the presented model expands available firn densification models, including an explicit representation of snow; (2) on the other hands, a simpler representation of the firn component results in a more straightforward integration in hydrological models. In the former case it is important to have as output a depth-density profile, while in the second an average firn density with which to estimate SWE may be enough.

Reflecting on the issue you pointed out, we decided to add a comparison between the two model versions that may relate the results of the two versions. It is in fact important, when adopting a simpler schematization, to be aware of the resulting approximation. For this purpose, we report here a comparison between the average firn density obtained from the single-layer version and the one obtained averaging the results of the multi-layer one (Fig.s RC1.1-2). In the figures, each group of bars represents a different run of the single-layer version of the model. On the x-axis it is reported the starting year of each simulation or equivalently the age of the oldest part of the firn column. The age of the most superficial part, or the ending year of the simulation, is different for the two cores: 2015 for CG15 and 2013 for KCC. The length of the bars is equal to the depth of the firn column obtained from the single-layer version or summing the depths of the individual layers of the multi-layer version. Dots, instead, represent the average density of the firn column, again as obtained from the single-layer version or averaging the density of each individual firn layer obtained with the multi-layer version, weighted for their depth.

[Figure]

**Figure RC1.1**: Firn density and depth obtained from the single-layer version of the snow-firn model and averaging the results of the multi-layer version, for ice core CG15, using the model for firn densification of Arnaud et al. (2000) (left panel) or Herron and Langway (1980) (right panel).

[Figure]

[Figure]

**Figure RC1.2**: Firn density and depth obtained from the single-layer version of the snow-firn model and averaging the results of the multi-layer version, for ice core KCC, using the model for firn densification of Arnaud et al. (2000) (left panel) or Herron and Langway (1980) (right panel).

Here the replies to the minor comments:

1. *L293-296  Why did you use the optimized values in the Japan site despite the studied field site being in Europe? If it is because Japan is the only place where the parameters of e could be adjusted, it should be written clearly. Also, it may lead to problems using a and e optimized at different locations in a single equation.*

A series of SWE or snow density with which to calibrate the parameter "e" was not available at Colle Gnifetti, but we may look for available SWE/snow density series in the Italian Alps and recalibrate the parameter in a site closer to the study site.

2. *L333-337, L377-389 From this result, the accuracy among several models are compared. Overall, the one that is calculated at high density seems to be suitable. Do you have any ideas to improve the model accuracy of the density profile more?*

In most of the ice cores reported in Fig. 4 of the manuscript it is possible to see a bend in the profile in correspondence of 20 - 30 m of depth. The reason of this could be a combination of ice flow and the upstream-effect, i.e. changes in snow accumulation upstream (P. Bohleber, personal communication, 26 April 2021).  These effects are not likely to be reproduced by a 1D model like the ones we used. However, several effects may influence the observed variability in depth-density profiles.

The application of the steady-state firn densification models to other cold alpine sites may possibly provide further information about the ability of Arnaud et al. (2000) or Herron and Langway (1980) models to reproduce depth-density profiles in sites with characteristics at the limit of the ones of polar regions. Thus, allowing to better separate the mismatch due to non 1D effects or non-stationarity from the mismatch due to the parameters calibration of firn densification models.

3. *L350-352, L468-470 In 1D simulations, I think the purpose of considering erosion is to avoid overestimation of the amount of new snow and underestimation of density. Do you have any verification of the amount of the erosion?*

At our knowledge, there are no measurements of wind erosion at Colle Gnifetti, therefore the only quantitative verification is the one coming from ice cores. Qualitatively, the erosion of winter snow is confirmed also by some temporary snow measurements performed in sites close by, like Colle Sesia.

4. *L355-366 In Fig. 4, the density profile was reproduced using a multi-layer model. On the other hand, what is the purpose of the validation of the density of one-layer model with averaged observed density. Also, I think quantitative comparison about the accuracy between one-layer and multi-layer model.*

The aim was to assess the accuracy of a simpler model in reproducing the average firn density in order to have a description of the snow water equivalent contained in firn and that will be potentially released under increasing temperatures. Following your comments, we added a comparison between the firn depth and density obtained directly from the single-layer version or averaging the results of the multi-layer one. This to answer the question, which approximation is introduced estimating the snow water equivalent of firn with a single-layer model for firn rather than a multi-layer one.

5. *L457-461 I understand that you lowered the temperature for snow melt to match the observed density. However, I wonder if it can change a fixed value in natural science such as the melting point in the simulation. It is still understandable if you are assuming the influence of unknown factors such as salinity. Please describe why you used melting point as an adjustment.*

By adjusting the melting temperature, we were trying to assess the uncertainty associated with a wrong estimation of the surface temperature of snow. The latter was, in fact, set equal to air temperature, while melting may occur also for air temperatures lower than zero or not occur for air temperatures higher than zero. The amount of melting varies greatly over the site; Mattea et al. (2021) estimated a mean annual melt between less than 1 cm w.e. yr -1 on the steepest slopes of the Signalkuppe that increases to 17 cm w.e. yr−1 at the saddle point and to about 23 cm w.e. yr−1 on the Zumsteinspitze slope. Since the snow-firn model estimates annual melt amounts closer to the lower limit obtained by Mattea et al. (2021) and the CG15 core is located near the saddle point, we tried to assess the influence of a possible wrong estimation of melting.

Mattea, Enrico, et al. "Firn changes at Colle Gnifetti revealed with a high-resolution process-based physical model approach." *The Cryosphere Discussions* (2021): 1-30.

6. *L482-484 I understand that a simple one-layer model is more convenient to integrate into a hydrological model, but it is a rough validation compared to the multi-layer model. If you are planning to use the one-layer model in the final integrated model, the multi-layer results should be fed back to the one-layer model in some way, otherwise the multi-layer comparison will be meaningless. Do you have any plans to connect the multi-layer results with the one-layer model in some way?*

In Fig.s RC1.1-2 we reported a comparison of the results of the two model versions that can provide information about the accuracy that we are losing moving from one version to the other, therefore giving additional information on the performances of the single-layer version.

---

## Author Comment (AC2)

Dear Referee #2,

thank you for your comments. We will reply to the general comments first and then we will address the specific ones.

Regarding the problem of parametrization, we will state more clearly in the manuscript that most of the fluxes are modeled based on empirical approaches, and we will rewrite the conclusions Section, following the remarks you presented in both the general and specific comments. We will also address the issues you pointed out regarding the structure and the use of English and we will check the presence of commas through the text.
The choice to present the selected values of some parameters in Section 3.4 while others in Section 2 was done in order to distinguish between site specific parameters (reported in Section 3.4) and the parameters whose value was not selected depending on the specific case study. If you believe it not to be clear or useful, we will reorganize Section 3.4 as you suggested.

Regarding the computation and representation of the depth-density profiles we will discuss it in the followings. In the single layer version, the overburden stress for firn densification was computed considering half of the firn column weight plus the whole weight of the snow column, in order to have an average densification rate. We will state this clearly in the manuscript. Regarding the multilayer version, we discretized the equation modelling a layer for each water year. For each layer, we then computed the overburden stress as the weight of all the layers above and half of the layer considered.

We apologize for the lack of clarity of the figures. We thought of a new representation of the results of the single layer version of the model, reported in Fig. RC2.1. In the panels of the figure, each group of bars represents a different run of the single-layer version of the model. On the x-axis it is reported the starting year of each simulation or equivalently the age of the oldest part of the firn column. The age of the most superficial part, or the ending year of the simulation, is different for the two cores: 2015 for CG15 and 2013 for KCC. The length of the bars is equal to the depth of the firn column obtained from the single-layer version or summing the depths of the individual layers of the multi-layer version. Dots, instead, represent the average density of the firn column, again as obtained from the single-layer version or averaging the density of each individual firn layer obtained with the multi-layer version, weighted for their depth. Whenever a snow layer becomes firn, the average density of the already existing firn column is changed depending on the density of the new added mass. This implies that average firn density, locally, may decrease with increasing simulation time, while the general trend will still be an increase in average density. We hope this helped in clarifying the issue, elsewhere we will provide further explanations of the results.

[Figure]

**Figure RC2.1**: Firn density and depth obtained from the single-layer version of the snow-firn model and observed, for ice core CG15 and KCC, using the model for firn densification of Arnaud et al. (2000) (AR) or Herron and Langway (1980) (HL).

We will also change the figures representing the results of the multi-layer version of the model, reporting a profile of depth versus density, as given in Fig RC2.2, instead of deposition year versus density. We are glad to receive any suggestions in order to further improve the representation of the results.

[Figure]

**Figure RC2.1**: Firn density and depth obtained from the multi-layer version of the snow-firn model and observed, for ice core CG15 and KCC, using the model for firn densification of Arnaud et al. (2000) (AR) or Herron and Langway (1980) (HL).

Here are the replies to the specific comments. Some comments are not reported, because we believed them not to need further discussions. They will, however, be integrated in the revised manuscript:

1.  *P2 L50: "momentum balance and rheological equations" → I don't really see any equations related to the momentum conservation (divσ+ ρg = 0) or firn rheology (S = σ+ pI = 2ηέ).*

    Following De Michele et al. (2013), to model snow densification, we have used the momentum balance equation in the integral form: σ + ρghs = 0, while concerning the rheology of the snow we have used the Maxwell law $\eta = \frac{\sigma}{\dot{\varepsilon}}$ where η is the coefficient of viscosity, and $\dot{\varepsilon}$ is the vertical viscous strain rate. The contribution of wind was then considered as proposed by Liston et al. (2007).
    Also the firn densification equation of Arnaud et al. (2000) is physically based. The first stage equation is obtained assuming linear-viscous boundary sliding, while the following equations were obtained assuming the constitutive equation of power law creep with a stress exponent varying from 3 to 1.

2. *P3 L84: "and metamorphosis of grains not driven by wind" → I don't think there is any representation of snow metamorphism in your model*

   You are right, the densification equation used models the evolution due to compaction, neglecting the metamorphism. We will correct it in the manuscript.

3. *P4 L113: Point (2) is not very clear, could you precise*

   We noticed that we reported incorrectly the point numbers, repeating two times point (2). We will correct this in the revised manuscript. If you are referring to the correct second point, the wind is measured at a given height from the ground, that is fixed when we have bare soil, but decreases when the snowpack is present. In the formula in which wind speed is modified so that it is referred to a specific height from the ground, we assumed the measure of the meteorological station to be referred always to 10 m above the surface. Regarding the third point, this assumption was introduced to model the role of melting in protecting snow from erosion. If snow melts and then freezes above the surface of the snowpack, it will create a hard layer that will prevent snow erosion. The presence of wet snow too makes the snow more difficult to be eroded. In the model, when snow melts, erosion flux is set to zero, independently of wind speed, until the deposition of new solid precipitation. We assumed, in fact, that, when a wet or icy snow layer is present, the snow below cannot be eroded.

   Regarding the last comment, after the list, we are able to compute the value of Q without running the snow-firn model, because it requires as input only the solid precipitation series, temperature and wind speed. Information about the snowpack height is not needed for point (2) and the amount of potential erodible snow can be inferred by solid precipitation alone if erosion after melting is not allowed i.e., point (3).

4. *P6 Eq (2b): I don't really understand how the last term of the right-hand side of this equation is treated as the firn is said to be impermeable. How do you transfer the remaining liquid water to the firn then?*

   The idea is that all the snowpack is moved into the firn layer at the end of each water year. Inside the firn, no water movement is then allowed, so that the water possibly present will remain confined in the firn pores.

5. *P8 Eq (4): In the text you are mentioning three densification regimes but there are four equations here. Could you clarified that point?*

   The last two equations belong to the same stage, i.e. after that close-off conditions are reached, when pores become isolated. The passage between the third and fourth equation is driven by the transition between cylindrical to spherical pores.

6. *P8 L196: It is not clear why the information regarding the transition density for HL is given here while the same information for AR is given at the end of Section 3.4. I would bring the latter back here (see Major Comments).*

The transition density in the model of Herron and Langway (1980) is given and it is not an adjustable parameter. On the contrary, in the model of Arnaud et al (2000), the transition density is one of the parameters whose value needs to be selected. Depending on the site, different values could be therefore chosen. As already reported above, we tried to separate the site-dependent parameters in the organization of the manuscript.

7. *P8 L201: "In this way the passage from snow to firn densification is driven by snow characteristics rather than snow age" → 1/ In this way, ... 2/ I don't really see where snow age appears as the driving factor for densification from Eq. (4) and (5).*

   We will reformulate it in the revised manuscript. Instead of moving from one densification equation to the other when the snowpack encounters the end of a water year, we linked the transition with the value of the average density of the layer. We believe it to be more accurate to use the information of snow density to guide the transition between the two densification equations.

8. *P8 L202: "corresponds to the average surface density" → Averaged over what ? Time ? Observations from cores ?*

   It is the average snow density, that we obtained extrapolating the most superficial part of depth-density profiles of ice cores. Steady-state firn densification models cannot be applied to the superficial snow where the metamorphism is more complex and significantly influenced by air temperature. The original model of Arnaud et al. (2000), for example, is applied only from a depth higher than 2 m. Therefore, in order to run them, it is necessary to specify a surface density. Different papers, studing the firn density at Colle Gnifetti, use different values of surface density, since the surface density is highly variable at the site.

9. *P10 Fig. 2: The map is a bit difficult to read. At least, I would remove the dotted lines (what is it ?) and the line made of crosses (boundary between Swiss and Italy ?) or add a legend to tell what they mean.*

   Thank you for your comment. We will change the basemap in order to make it clearer.

10. *P11 Fig. 3: It would be great to have a box on the background map to know exactly to what region the zoom-in of the lower right corner corresponds. Also, the fact that it is a zoom-in should be precised in the caption. You can also mention that CM appears on Fig. 2 as well, which will help the reader to localize the study area on this map*

    Thank you for your suggestion. We will add the box and rewrite the caption.

11. *P13 L283: "smoothing the snow depth with a moving average whose window size was calibrated" → Not very clear, could you clarified ?*

    Hourly snow depth series cannot be used to infer solid precipitation series by simply computing hourly differences; this because of the fluctuations of snow depth measurements due to air temperature variations, that introduce positive differences not related with solid precipitation events. We therefore processed the series of positive differences of hourly snow depth, applying a moving average to the series. To chose the best window for the moving average, for each water year, we selected several window sizes and we calibrated *a* for each of

them. Among all the windows, for each year, we chose the one with the best value of the objective function in calibration.

12. *P14 L296-301: The procedure you are following to calibrate a is not very clear. First, I don't understand why the wind contribution is not accounted for as it should explain part of the ablation, the other part being due to melting. Then, if I understand well, you are calculating one optimal value of a per hydrological year, and then, for each thus obtained value of a, you compare observed snow depth to snow depth modelled for all the hydrological years except the one from which the considered value of a has been derived. Is that correct? If yes, I think that this procedure makes sense, but what I find surprising is that you don't use these comparisons between model outputs and observations to set the final value of a but you simply take the median of all the a you have calculated. Why is that?*

The parameter *a* was calibrated to a lower altitude site, using data of a meteorological station located in a position sheltered from wind. For this reason, wind effect was not taken into account. Regarding the second point, you understood correctly. The comparisons were used to compute the performances of the model in validation, but in order to select the final value of the parameter *a*, as done also by De Michele et al (2013), we did not consider which parameter' value had better performances in validation.

13. *L304-306:"Its value is in fact chosen in order to have a continuous densification rate between first and second stage of densification. For each of the available ice cores, with the exception of CG11, we computed the parameter γ' running AR in a steady-state condition (Bader, 1954) using the mean accumulation reported in Table 2." → In other words, you are calibrating γ', right?*

You are correct, we will change this in the revised manuscript.

14. *P14 L308-312: The way you are setting the value of ρD for the various cores sounds a bit random as presented here*

In Fig. 4, the purpose was simply to evaluate the performances of steady-state models, therefore we selected a value of surface density already used in literature to study firn at Colle Gnifetti. For the two cores, used to compare non steady-state results, we checked if the selected value was coherent by extrapolating the last part of the depth-density profile of firn. We consequently kept the value of 360 kg/m$^3$ for KCC and we estimated a new value, from the extrapolation, for CG15. Moved by your comment, we report here a comparison between the results of the snow-firn model for CG15 ice core with $\rho D = 360$ kg/m$^3$ (one of the two values tested by Licciulli et al (2020)) or $\rho D = 470$ kg/m$^3$, the value used in the manuscript.

[Figure]

[Figure]

**Figure RC2.3**: Firn density and depth obtained from the multi-layer version of the snow-firn model and observed, for ice core CG15. In the left panel, using the model of firn densification of Arnaud et al. (2000) with a transition density between snow and firn of 470 kg/m3 (AR470) or 360 kg/m3 (AR360). In the right panel, same as left panel but for Herron and Langway (1980) firn densification model.

15. P14 L319: The parameterization of SSA given here is not consistent with the one given in Appendix A1.

    We are using two different parametrizations for SSA depending on the context. The one in the appendix is used to estimate the grain size of the whole snowpack. In order to compute the grain size in wind erosion formula, we adopted a parametrization proposed for recent snow, since we believe it to be more representative of the grain size of the snow mobilized by wind.

16. *P15 L323: Why does the unit of NSE, RMSE and MBE is different from the original unit of a?*

    Calibration and validation were performed using snow depth in the objective function so the error measures have unit of a depth.

17. *P15 L324: What is the SNOTEL network?*

    SNOTEL (Snow Telemetry) is a network of automated stations, located in mountain basins of western U.S., operated by the Natural Resources Conservation Service (NRCS). They collect data of precipitation, snowpack characteristics, temperature and other meteorological variables. We will clarify this in the revised version of the manuscript.

18. *P15 L329-331: I have the feeling that a re-calibration of parameter a would be required if the value of e is changed to be consistent. Is it not the case?*

    Thank you for your remark. We will remove the sensitivity analysis of the parameter *e*.

19. *P18 Fig. 6: For the lower right Figure, I think what is represented is 100 × (Solid precip. - Eroded snow)/ Solid precip., am I correct ? If yes, you could write it explicitly.*

    Yes, we will write it explicitly.

20. *P19 Fig. 7: I am not sure this Fig. brings relevant additional information.*

    We will remove it in the revised manuscript.

21. *P20 L386: Why comparing HL and AR between each other and not to observations ? In addition, in its current state, this sentence seems to countradict the one just before.*

    In Fig. 4 we report the performances of the steady-state firn densification models as they were proposed by the respective Authors, without introducing novelties. Nevertheless, it is important to understand their performances at Colle Gnifetti and also how the two models behave, before combining them inside the proposed snow-firn model. In this context, we included the comment about the differences between the two models. Comparing the two models, we can see that the profile obtained with AR is generally above the one obtained with HL before $D_0$ and below after $D_0$. This may give information about a tendency of one of the two models to overestimate/underestimate the density.

22. *P21 L408-409: How do we compare cm to kg m−2 yr−1 ? More broadly speaking, a choice has been made to give all results in terms of accumulation in kg m−2 yr−1, but I think that m SWE is often easier to apprehend. But that's only my opinion.*

    This a second submission of the manuscript. The first TC Editor, that handled it, suggested us to change all results from m SWE to kg m−2 yr−1.

23. *P22 Fig. 10: The choice of the colors is not ideal.*

    Thanks for the remark. We will change them.

24. *P23 L442:"The annual behaviour" → You mean the inter-annual variability, don't you ?*

    Yes, we do. We will correct it.

25. *P23 L456-457:"along with the snow redistribution due to gravity" → What do you mean ?*

    The spatial pattern of accumulation in the site may also be due to the movement of deposited snow along the saddle due to gravity. We will clarify this point in the revised manuscript.

26. *P23 L460-461:"We can instead see an improvement in the ability of the model to follow year-to-year variations in the period 2008-2014" → It is not very obvious.*

Thank you for your remark. We looked at the results of Fig. 11, plotted as a density versus depth, instead of deposition year, and we saw no improvements by changing $T_t$. We will remove this discussion from the manuscript.

27. *P24 L474:"Results in Fig. 9, on the contrary, show a better and more robust performance of the model with" → one cannot say that it is robust when it changes depending on the core being considered*

    Thank you for your comment, we will change it. We will also change Fig. 9, substituting it with the profiles of density versus depth, reported above, and we will comment the performances of the model based on the new figures.

28. *P24 L485:"to asses the influence of meteorological variables on snow and firn characteristics." → (1) to assess; (2) How this could be helpful ?*

    As stated before, the model links meteorological variables, used as inputs, to the firn characteristics. It can be therefore run with simulated meteorological series, as input, in order to understand how firn responds to changes in temperature or precipitation.

29. *P25 L486:"therefore moving the boundary of the model from surface accumulation and density to hourly meteorological series" → This is an interesting way of formulating what you have been doing in this paper. I think a similar formulation of the problematic you are addressing would be welcomed in the introduction as well.*

    Thank you for your suggestion. We will introduce this formulation of the problem in the Introduction as well.

---

## Author Response (AR1)

**Dear Editor, Dear Referees,**

thank you again for your evaluation of our manuscript. Working on the revised version of the manuscript and reflecting further on your comments, we decided to perform some changes with respect to what already stated in the replies. In particular, we would like to make additional comments to the following points of the evaluation by Referee #2:

• P8 Eq (4) Top rhs member: Remove the Pa (if that stands for Pascal, pressures should naturally be given in Pascals) and check wether the 10 4 has to stay or not

The equation needs the maximum between the overburden stress and 0.1 bar, i.e. 10000 Pa. In order to better model the densification of the most superficial part of firn, Bréant et al. (2017) imposed a fixed constant pressure of 0.1 bar, that should approximate the pressure at 2-3 m depth, when overburden stresses are lower. We reported the unit of measure because we believe it to be clearer, since this value can be assimilated to the reported values of constants.

• P14 L296-301: The procedure you are following to calibrate a is not very clear. First, I don't understand why the wind contribution is not accounted for as it should explain part of the ablation, the other part being due to melting. Then, if I understand well, you are calculating one optimal value of a per hydrological year, and then, for each thus obtained value of a, you compare observed snow depth to snow depth modelled for all the hydrological years except the one from which the considered value of a has been derived. Is that correct ? If yes, I think that this procedure makes sense, but what I find surprising is that you don't use these comparisons between model outputs and observations to set the final value of a but you simply take the median of all the a you have calculated. Why is that ?

For the calibration of the two parameters, *a* and *e*, that was performed in a different site in the revised manuscript, we accounted for wind contribution, running the snow model with the addition of wind erosion and the influence of wind on snow density. We also changed the adopted procedure with which we selected the optimum parameters, as reported in the revised manuscript. In particular, due to the low amount of available snow water equivalent data, we calibrated the two parameters considering together all the years, instead of performing a different calibration for each year. We then performed the validation on a site closer to Colle Gnifetti, in order to check the performance of the selected parameters and their transferability. The choice of the calibration and validation station was made so as to calibrate the parameters in the site where SWE measurements were performed closer to the meteorological station. This because the solid precipitation input series was reconstructed from the snow depth series measured at the meteorological station; hence, having measured SWE data in a site characterized by different snow depths is likely to introduce more errors in the model.

P14 L308-312: The way you are setting the value of  $\rho$  D for the various cores sounds a bit random as presented here.

In the revised version, we tested three different values of  $\rho D$  looking for values already adopted in the literature at CG.

---

## Referee Report (RR1)

**General comments**

Overall, the authors of 'A local model of snow-firn dynamics and application to Colle Gnifetti site' have taken my comments into account and have accomplished substantial work which has considerably improved the quality of the manuscript. In particular, I highly appreciate that the calibration procedure for parameters $a$ and $e$ was revised, with calibration performed at a station where wind erosion as a significative effect on total accumulation. I also think that the presentation of results regarding the evolution of the modeled firn density, both in the single-layer and multi-layers versions, is now much clearer. This is for a large part due to the new Figs. 7 and 8 which, from my point of view, are much easier to interpret than the ones on which the previous version of the manuscript was relying.

Regarding the structure of the paper, I like the fact that there is now a Section 'Calibration of model's parameter' and another one 'Site specific parameters'. The manuscript is much better organized this way. Many commas have been added, but some are still missing here and there.

Overall, I think the manuscript is almost ready for publication, I have simply a last handful of minor comments that are listed below. Note that page and line numbers refer to the version of the manuscript with tracked changes.

**Specific comments**

P2-3 L58-59: This new sentence lacks of clarity and would deserve to be reformulated.

P4 L92: Change slightly the sentence to avoid the repetition of 'compaction'. I think it is correct to use 'settlement' instead of the second occurence of 'compaction'.

Eq (1d) p4 and Eq (3f) p6 L23-25: Following your answer to my comments about the momentum balance and firn rheology, I checked the paper of Di Michele et al. (2013) to see how expressions of Eq (1d) and (3f) were derived from mass/momentum conservation and Maxwell law. It turns out that in their formula the density in the factor before the exponential is raised to the square, whereas it is not in your expression. Please double-check this point, both in the manuscript and in the code of the model.

P8 L191: 'that may be assimilated to the conditions' $\rightarrow$ 'assimilated' sounds a bit too strong to me. What about 'that resemble conditions' ?

P8 Eq4 Top rhs member: I keep thinking that the 'Pa' should be removed as $P$ is already given in Pa as you precise it at l. 195. Therefore, for reason of homogeneity of the equation, the second member of the max operator is naturally also given in Pa. This is only a formality and I leave the decision to the editor.

P8 L208: One comment: I have the feeling that because the density increases non-linearly with depth, there are more mass in the lower half of the firn column than in the upper half. Therefore, considering the overburden pressure of the snowpack plus the upper half of the firn column is probably not a very good estimate of an 'averaged pressure' that would apply to the whole firn column. And therefore, calculating the densification rate from this overburden pressure value does not necessarily give an 'averaged densification rate of the firn column'. However, one could probably argue that this is somewhat counterbalanced by the fact that firn deforms less readily at higher densities.

P12 L292: There is an inconsistency here as you are saying just above that for gaps longer than 24h, you adopt the long-term lapse rate approach and not the MicroMet procedure.

P14 L314-318: Despite the effort you have made to make it clearer which I acknowledge, I stil have a hard time to understand what you have been doing here. But if the editor finds it clear enought, then you can leave it as it is.

P16 L380: "the three NSE" $\rightarrow$ It took me a little while to understand that the three NSE were refering to the combined NSE as well as the one for snow depth only and the one for SWE only. Maybe it is worth to repeat it.

P18 L406: The quantities you are representing as monthly box plots are not listed in the text in the same order as they appear in Fig. 6, which is somewhat confusing, especially since the way yo are referring to them in the text

does not correspond to the titles of the subplots.

Fig.7 p21 and Fig.8 p22: It is really great to have made these new figures that are much clearer than the older ones. However, I have one small comment: in Fig. 7 rows are for the densification model and columns for the considered core, while it is the other way round in Fig. 8. I think it would be better to have it the same in both Figs.

P24 L472-480: Are you talking about the change in densification rate which is very obvious in, e.g. cores CG03,CC and KCI ? If yes, is it not rather due to the switch toward another densification regime at high density that could correspond to the third densification stage that you mentionned in p.7 ?

P25 L516: This sentence needs to be checked for meaning. In addition, I would insist on the fact that this sentence refer only to the configuration of CG or similar configurations, i.e. that *in general* higher temperatures are not expected to lead to higher snow accumulation as I feel that this sentence, in its current formulation, is suggesting.

P26 L534: reasonable

---

## Author Response (AR2)

Dear Editor, dear Referees,

thank you for your comments. Here are our replies.

Referee #1

Thank you for your suggestion of integrating the results of the multi-layer version of the model in order to improve the performance of the single layer one. We will address it in the future.

Referee #2

- *P2-3 L58-59: This new sentence lacks of clarity and would deserve to be reformulated.*

  Thank you. We reformulated it (ll 55-57).

- *P4 L92: Change slightly the sentence to avoid the repetition of 'compaction'. I think it is correct to use 'settlement' instead of the second occurence of 'compaction'.*

  Done.

- *Eq (1d) p4 and Eq (3f) p6 L23-25: Following your answer to my comments about the momentum balance and firn rheology, I checked the paper of Di Michele et al. (2013) to see how expressions of Eq (1d) and (3f) were derived from mass/momentum conservation and Maxwell law. It turns out that in their formula the density in the factor before the exponential is raised to the square, whereas it is not in your expression. Please double-check this point, both in the manuscript and in the code of the model.*

  For the dry snow densification, to take into account wind effects, we used the equation proposed by Liston (2007). In the equation, strains are still governed by the same coefficient of viscosity, but the stress driving strains is parameterized as a function of wind speed so in the equation there is not the term ($\rho_s * h_s$).

- *P8 L191: 'that may be assimilated to the conditions' → 'assimilated' sounds a bit too strong to me. What about 'that resemble conditions'?*

  We changed it as you suggested.

- *P8 Eq4 Top rhs member: I keep thinking that the 'Pa' should be removed as P is already given in Pa as you precise it at l. 195. Therefore, for reason of homogeneity of the equation, the second member of the max operator is naturally also given in Pa. This is only a formality and I leave the decision to the editor.*

  We do not have a specific interest in keeping the unit 'Pa', we simply believed it not to be clear otherwise. But, since from your comment I understand this is not the case, we removed it in the revised version as suggested.

- *P8 L208: One comment: I have the feeling that because the density increases non-linearly with depth, there are more mass in the lower half of the firn column than in the upper half. Therefore, considering the overburden pressure of the snowpack plus the upper half of the firn*

*column is probably not a very good estimate of an 'averaged pressure' that would apply to the whole firn column. And therefore, calculating the densification rate from this overburden pressure value does not necessarily give an 'averaged densification rate of the firn column'. However, one could probably argue that this is somewhat counterbalanced by the fact that firn deforms less readily at higher densities.*

Thank you for your comment. We will explore it in future developments of the model.

- *P12 L292: There is an inconsistency here as you are saying just above that for gaps longer than 24h, you adopt the long-term lapse rate approach and not the MicroMet procedure.*

The MicroMet procedure consists of three sub-methods depending on the gap size, where the last one was substituted in our work with another approach. Nevertheless, we preferred to report the complete MicroMet procedure. To avoid confusion, we added a sentence (l.284) to make it clearer that the last step was not used here.

- *P14 L314-318: Despite the effort you have made to make it clearer which I acknowledge, I stil have a hard time to understand what you have been doing here. But if the editor finds it clear enought, then you can leave it as it is.*

The procedure used is very similar to the one reported in the paper by Avanzi et al. (2014), with the exception reported in the manuscript. For this reason, we did not want to dedicate too much space for its explanation, to not extend the manuscript with information already published. However, we will explain it with more details if the editor believes it necessary.

- *P16 L380: "the three NSE" → It took me a little while to understand that the three NSE were refering to the combined NSE as well as the one for snow depth only and the one for SWE only. Maybe it is worth to repeat it.*

Done

- *P18 L406: The quantities you are representing as monthly box plots are not listed in the text in the same order as they appear in Fig. 6, which is somewhat confusing, especially since the way yo are referring to them in the text does not correspond to the titles of the subplots.*

Thank you for noticing it. We changed the order in which they are listed in the text so that now it matches the one in the figure.

- *Fig.7 p21 and Fig.8 p22: It is really great to have made these new figures that are much clearer than the older ones. However, I have one small comment: in Fig. 7 rows are for the densification model and columns for the considered core, while it is the other way round in Fig. 8. I think it would be better to have it the same in both Figs.*

Done

- *P24 L472-480: Are you talking about the change in densification rate which is very obvious in, e.g. cores CG03,CC and KCI ? If yes, is it not rather due to the switch toward another densification regime at high density that could correspond to the third densification stage that you mentionned in p.7 ?*

Taking as an example CG03 ice core, I am referring to the change occurring at a density of around 700 kg/m$^3$. I believe this not to be the case, for two main reasons. First, I expect the close-off density, i.e. the passage between second and third stage, to be associated with a lower density. Luthi and Funk (2000), for example, for Colle Gnifetti, assumed it to be between 780 and 917 kg/m$^3$. Secondly, when the close-off density is reached, densification rate decreases, because it is now associated only to the compression of the bubbles of air. Hence, the third stage is characterized by a steeper profile of density with depth. On the contrary, the change, pointed out in the manuscript, results in a profile that is more horizontal, and therefore in a greater variation of density with depth with respect to a smaller depth.

- *P25 L516: This sentence needs to be checked for meaning. In addition, I would insist on the fact that this sentence refer only to the configuration of CG or similar configurations, i.e. that in general higher temperatures are not expected to lead to higher snow accumulation as I feel that this sentence, in its current formulation, is suggesting.*

We rephrased the sentence, and we hope that in this way we made it clear that we are discussing the specific response of Colle Gnifetti under climate changes (ll 459-462).

- *P26 L534: reasonable*

Done.